# Prevalence and risk factors of intestinal protozoal infections among patients in Malaysia: A systematic review and meta-analysis

Nor Shazlina Mizan[1], Seok Mui Wang[2,3,4], Hasnah Ma'amor[5], Hassanain Al-Talib[2*]

**1** Faculty of Medicine, Institute for Medical and Molecular Biotechnology (IMMB), Universiti Teknologi MARA (UiTM), Sungai Buloh, Selangor, Malaysia, **2** Department of Medical Microbiology and Parasitology, Faculty of Medicine, Universiti Teknologi MARA (UiTM), Sungai Buloh, Selangor, Malaysia, **3** Cardiovascular Advancement and Research Excellence Institute (CARE Institute), Universiti Teknologi MARA (UiTM), Sungai Buloh, Selangor, Malaysia, **4** Non-Destructive Biomedical and Pharmaceutical Research Center, Smart Manufacturing Research Institute (SMRI), Universiti Teknologi MARA (UiTM), Puncak Alam, Selangor, Malaysia, **5** Sector for Evidence-Based Healthcare, National Institutes of Health, Ministry of Health, Shah Alam, Selangor, Malaysia

\* hassanain@uitm.edu.my

## Abstract

### Backgrounds

Intestinal protozoan infections (IPI), including *Entamoeba spp., Giardia lamblia* and *Cryptosporidium spp.*, are common in diarrhoea patients in Malaysia. These parasites are primarily transmitted through contaminated food and water sources and pose a significant public health concern. *Entamoeba spp.,* causes amoebiasis, which is characterised by severe diarrhoea with bloody stools, while *Giardia spp.,* causes giardiasis, which is characterised by watery diarrhoea, abdominal pain and flatulence. *Cryptosporidium spp.* causes cryptosporidiosis, which is particularly severe in immunocompromised individuals. Despite efforts to improve water quality, sanitation, hygiene, and surveillance, limited epidemiological data prevent a clear understanding of the prevalence of intestinal protozoa and response to treatment. This systematic review and meta-analysis aimed to estimate prevalence, identify risk factors, and evaluate detection methods.

### Methods

A comprehensive literature search was carried out in the following databases: Scopus, Google Scholar, Web of Science, PubMed and Cochrane Library. The review was conducted according to the Preferred Reporting Items for Systematic Reviews and Meta-Analyses (PRISMA) 2020 guidance. Based on the random effects model, results were reported as proportions with 95% confidence intervals (CI). Subgroup analyses were performed based on detection methods, target populations and regions.

**Data availability statement:** All relevant data are within the manuscript and its Supporting Information files.

**Funding:** This research received fund from Universiti Teknologi MARA (UiTM), 600-RMC/PRGS 5/3 (004/2024) The funders had no role in study design, data collection and analysis, decision to publish, or preparation of the manuscript.

**Competing interests:** The authors have declared that no competing interests exist.

## Results

A total of 103 articles were found on the prevalence and risk factors of IPI in Malaysia. After removing duplicates and screening for eligibility, 49 studies were included in this meta-analysis. The overall pooled prevalence of IPI in Malaysia was 24% (95% CI: 0.17.0, 0.29). with $I^2 = 98.94\%$, (P < 0.001). Among the identified protozoa, *Entamoeba spp.* had the highest prevalence at 18% (95% CI: 0.12, 0.24), followed by *G. lamblia* at 11% (95% CI: 0.08, 0.14) and *Cryptosporidium spp.* at 9% (95% CI: 0.03, 0.14). A subgroup analysis revealed that Kelantan and Perak state have the highest prevalence of 39% and 29% while Selangor and Kuala Lumpur reported the lowest (13.6%). The highest prevalence was observed in the indigenous communities (27%), followed by the local communities that mainly comes from rural area (23%). According to a meta-analysis of ten risk factors, the pooled prevalence of protozoal intestinal infections was significantly higher (between 38% and 52%) in children under 15 years of age, in males, in those with low income or no formal education, and in those exposed to untreated water, poor sanitation or unhygienic practises. A high level of heterogeneity was observed ($I^2 > 98\%$), reflecting substantial variability across the included studies.

## Conclusion

This review provides valuable insights into the epidemiology of protozoal intestinal infections in Malaysia. The high pooled prevalence of 24% underscores a substantial and ongoing burden of intestinal protozoal infections in Malaysia. The pooled prevalence should be interpreted with caution due to high heterogeneity, as the findings may not be generalizable to all settings. These findings support the development of evidence-based interventions to reduce the impact of these infections. Targeted screening, improved diagnostics, better access to clean water and sanitation, and health education for vulnerable groups are essential to reduce the burden of intestinal protozoa and strengthen national control measures.

## 1. Background

Intestinal protozoal infections (IPI) are the most common type of parasitic infection and a serious global health concern [1]. In developing countries, these infections are widespread and mainly affect children living in the most vulnerable populations [2], including the most endemic regions such as sub-Saharan Africa, Southeast Asia, China, South India and South America [3]. It is estimated that 3.5 billion people are affected and approximately 450 million people currently suffer from IPI [4]. If AIDS is synonymous with promiscuity, then IPI are synonymous with poverty, as these infections rarely cause symptoms in humans. Most epidemiological studies have usually focused on their prevalence [5].

The most frequently reported cases of infection with intestinal protozoa are *Entamoeba histolytica, Giardia lamblia* and *Cryptosporidium parvum*. These protozoa are transmitted by various routes, e.g., oral-faecal, i.e., indirect or direct contact, human-to-human, zoonotic, waterborne, foodborne (*E. histolytica* and *G. lamblia*) and airborne (*Cryptosporidium* s*pp*.) [6]. The most common symptoms of an IPI are nausea and watery diarrhoea. This is caused by the release of enterotoxin, which is often accompanied by inflammation in the stomach, small intestine and large intestine [7]. The causes of infectious diarrhoea are difficult to determine as most enteric pathogens cause similar symptoms leading to multiple infectious agents causing acute gastroenteritis (AGE). In addition, contamination can occur through food, water, the environment, or animals. Therefore, it is difficult to establish an accurate and rapid epidemiological analysis [8,9].

Every year, almost 50 million cases of invasive *E. histolytica* diseases occur, causing up to 100,000 deaths. Entamoeba infections affect 50% of the world's population, mainly in areas of Central and South America, Africa and Asia, and up to 25% in certain regions of underdeveloped tropical countries [10]. Same cases of giardiasis are caused by the protozoan parasite *G. lamblia* and it is the predominant intestinal parasitic infection that has been frequently documented in the United States and the United Kingdom. It is estimated that there are approximately 200 million infections annually in Africa, Asia, and Latin America. Furthermore, the prevalence of this infection appears to be particularly high in tropical countries. It has been documented in 13% of children with diarrhoea in India and 7.3% in Thailand. [11,12]. Overall, these findings emphasise the significant burden of these parasitic infections, particularly in underdeveloped tropical countries and regions in Central and South America, Africa and Asia. Efforts to prevent and control these infections should be prioritised, with a focus on improving sanitation and hygiene practices in affected areas.

Given that IPI remain a significant global health concern, Malaysia is a valuable setting for investigating IPI due to its multi-ethnic population and the coexistence of rapidly urbanising regions with remote rural communities [2]. The risk and prevalence of infections in these environments are characterised by persistent inequalities in access to healthcare, water quality, and sanitation [13]. In Malaysia, helminths and protozoa are the most common types of parasites that cause intestinal protozoan infection [5]. Diarrhoea is the leading cause of death in children under the age of five, with a reported mortality rate of 0.8% in 2019 [14]. In addition, the prevalence of diarrhoea in children under five years of age was found to be 4.4% [9]. In addition, numerous studies have demonstrated a significant incidence of amoebiasis with a prevalence of 1% to 14% and giardiasis with a prevalence of 2% to 19.4% [5,15,16]. *Cryptosporidium spp*. was found in about 4.3% of children with diarrhoea [17]. In addition, a study conducted in Malaysia revealed an increased rate of cryptosporidiosis among intravenous drug users with HIV-positive individuals, although the carriers were asymptomatic [18]. Meanwhile, *Entamoebiasis* was found in 10.2% of the Orang Asli settlement [19]. Previous studies in indigenous populations have documented varying prevalence rates, ranging from 9.4% to 18.5%. However, in this study, microscopic examination was used as the method, unable to distinguish different types of infections caused by *E. histolytica, E. dispar* and *E. moshkovskii* [20].

Currently, there is limited epidemiological data in Malaysia on the prevalence of IPI in immunocompromised individuals, especially those who are HIV. For example, Asma et al (2011) investigated the association between cryptosporidiosis and IPI in HIV-infected immunocompromised individuals [21]. However, most of these studies focused on hospitalised patients, with one study specifically investigating drug addicts who were also HIV-positive. The prevalence of intestinal parasitic infections (IPI) among prison inmates in Malaysia revealed that HIV-positive inmates had a slightly higher rate of IPI with a prevalence of 27.5% compared to HIV-negative inmates with 25.8% [22]. Malaysia also faces a significant burden of intestinal parasites, particularly in children under five and certain vulnerable populations [23,24]. The emergence and widespread of IPI in Malaysia are significant events in the epidemiology of infectious diseases. Unfortunately, there is limited epidemiological survey of these infections in Malaysia as only sporadic studies are available. Despite these limited studies, the overall prevalence of intestinal protozoa in clinical samples and their susceptibility to specific treatments in the Malaysian population remains unknown. To address this knowledge gap, we conducted a systematic review and meta-analysis to estimate the overall prevalence of IPI and identify associated risk factors in patients in Malaysia. This study builds on our previously published protocol and applies the described methodology to summarize current evidence,

assess regional and diagnostic differences and provide updated, evidence-based prevalence estimates [25]. The findings will inform public health strategies and improve surveillance and control measures for IPI in the Malaysian context [26].

## 2. Methods

### 2.1. Search strategy

A comprehensive literature search of the following databases from 15 years from 2010 to 2024 was conducted by the first author (NSM) according to the recommendations of the Preferred Reporting Items for Systematic Reviews and Meta-Analyses (PRISMA) 2020 statement in supplementary material (S1 File). Relevant studies were identified by a systematic search in the databases PubMed/MEDLINE, Scopus, Google Scholar, Web of Science and the Cochrane Library. The results of each search were published in EndNote Volume X. (Clarivate Analytics, PA, USA). Medical subject headings and the keywords were used in the search engine and matched between the first author (NSM), the lead author (HAT) and another lead author (WSM). Different keywords were selected, and the search was performed using "AND" and "OR" in the search section of the databases (Table 1). The complete search strategy used for all databases can be found in S1 Appendix. The reference lists of retrieved studies were used to identify additional studies and were selected based on the inclusion criteria of the systematic review. The review registered in PROSPERO: registration number CRD42023456199. Studies were selected if they met the following criteria: Primary studies that were conducted in Malaysia, and reported on the type of prevalence of intestinal protozoal infection, risk factor, and type of clinical samples collected from patients. Studies were excluded if: Studies not conducted in Malaysia, samples specimens isolated from food, animals, and biotic components (water, air, light, soil and temperature), and the full text was not available. In addition, we also excluded case reports, reviews, or conference abstracts. The selection or exclusion of studies is based on the criteria proposed in Table 2 and recorded the selection process in sufficient detail to complete a PRISMA flowchart diagram shown in Fig 1. The list of included and excluded studied can be found in S2 and S3 Appendix. The limited availability of human-related studies from East Malaysia likely stems from ethical, logistical, and practical constraints in conducting sampling [27]. Numerous studies from this region were excluded as they focused on animal hosts or environmental contaminants rather than human infections, or did not examine the protozoa of interest

### 2.2. Data extraction

Data extraction was performed by the main author (NSM) and cross-checked by (HAT, WSM, and NHM). The data sheet of a screening checklist consists of study details: author, year of publication, period of the study, location where the study was conducted, study characteristics (sample size, mean age and age range of participants), measures of exposure (socioeconomic status, geographic area, population characteristics), outcome (number of species identified, identified species and the total number of cases), and the association between diagnostic methods and the presence of intestinal protozoa status (prevalence, odds/ratios, odds/relative risks or hazard ratios) with associated variability (standard deviations or errors [SD or SE] and 95% confidence intervals [95% CI]). All major limitations of the study were identified and

**Table 1. Search strategy.**

| 1 | "Epidemiology"[Mesh] OR Prevalence*[tw] OR Incidence*[tw] OR Trend*[tw] OR Rate*[tw] OR "Epidemiological Stud*"[tw] OR "Cross-Sectional Stud*"[tw] OR "Observational Study*"[tw] |
|---|---|
| 2 | "Risk Factors"[Mesh] OR Risk factor*[tw] OR Mortalit*[tw] |
| 3 | "Protozoan Infections"[Mesh] OR "Intestinal Protozoa*"[tw] OR "Enteric protozoa*"[tw] OR Giardiasis[tw] OR Cryptosporidiosis[tw] OR Entamoebiasis[tw] OR Entamoeba[tw] OR Giardia[tw] OR Cryptosporidium[tw] |
| 4 | "Malaysia"[Mesh] OR "Peninsular Malaysia" [tw], "East Malaysia" [tw] |
| 5 | #1 AND/OR #2 AND #3 AND #4 |

**Table 2. Inclusion and exclusion criteria.**

| Category | Included | Excluded |
|---|---|---|
| Population | Both asymptomatic and symptomatic cases from different socioeconomic backgrounds, including all age groups, genders, ethnicities and geographical locations in Malaysia (including non-Malaysians) identified based on laboratory tests, medical diagnoses or medical records. | Both asymptomatic and symptomatic cases conducted outside Malaysia or in non-Malaysian settings, including cases from mixed communities, outpatient, inpatient and residential facilities, regardless of age, gender, ethnicity or non-Malaysian. |
| Intervention/exposure | Studies of the three most common intestinal parasites (*Entamoeba histolytica*, *Giardia lamblia* and *Cryptosporidium parvum*) carried out using one of the following methods: microscopic examination, immunoassay technique and molecular methods. | Studies excluding these three intestinal protozoa (*Entamoeba histolytica*, *Giardia lamblia* and *Cryptosporidium parvum*) and other intestinal protozoan parasites not associated with this study or using other detection methods as mentioned. |
| Study design | Quantitative studies, cross-sectional studies, cohort and case control studies. | Case reports, reviews and studies without original data. |
| Outcome | Diarrhoea and/or presence intestinal protozoa. | Non-enteric infection and studies without detection method mentioned. |

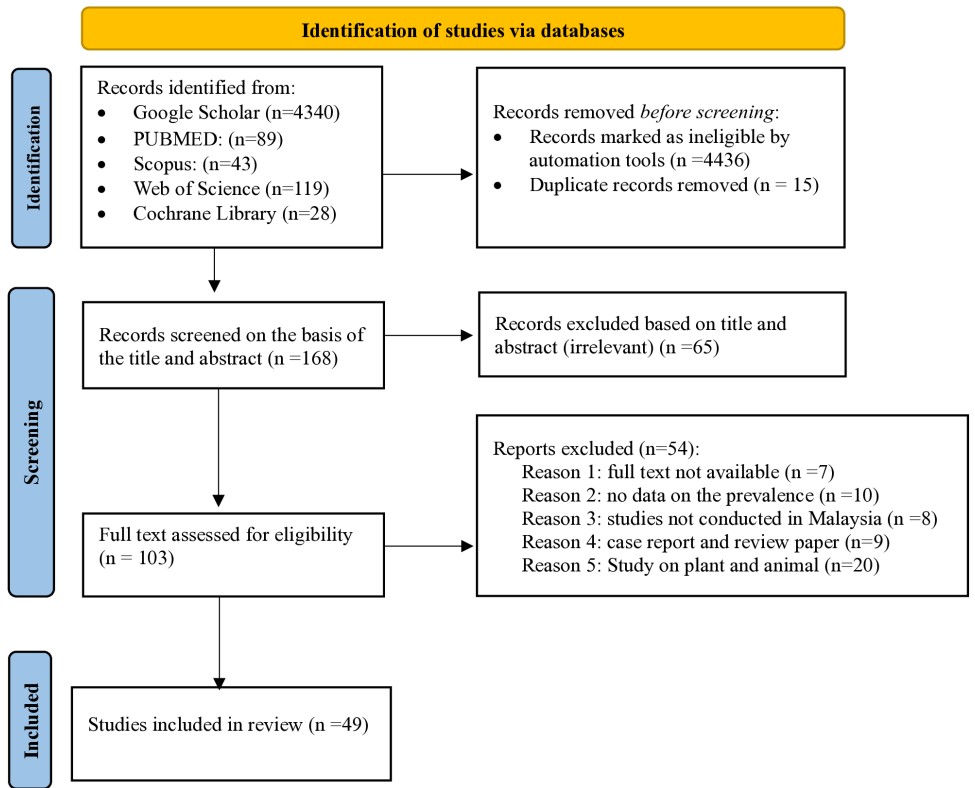

**Fig 1. PRISMA flow chart.**

further relevant information was requested from the corresponding author(s). For studies with missing or incomplete data, we attempted to contact the corresponding authors for clarification. If we did not receive a response, the available data were used as indicated. Studies with critical missing information that could not be clarified were excluded from the quantitative analysis.

## 2.3. Quality assessment

Risk of bias (quality assessment) was assessed using the Joanna Briggs Institute critical appraisal tools [20]. The lead author and two independent reviewers assessed each study individually at both outcome and study level to provide an overall assessment of risk of bias. Each reviewer rated each study as "yes", "no", "unclear" or "not applicable" in terms of bias (S5 Appendix). This tool ensured a consistent and balanced assessment of the methodological quality of all studies included in the review, taking into account the variability between experimental and observational study designs.

## 2.4. Data synthesis

All data were analysed using STATA software (version 17.0; William Gould) and Open Meta Analyst (CEBM, University of Oxford). Results were summarised in tabular form, with key outcomes, including gender, age, region and other variables, presented to explore relationships between outcomes and risk factors. The heterogeneity index ($I^2$) indicates considerable variability among the included studies, probably due to differences in study design, population characteristics, geographical locations and diagnostic methods. This reflects the different methods used to identify and report intestinal protozoal infections. A random effects model was used to calculate the overall prevalence, taking into account both clinical and epidemiological heterogeneity. The results were expressed as proportions and 95% confidence intervals. A narrative analysis summarised the key characteristics and quality of publications, and each study was assessed for risk of bias.

## 3. Results

A total of 4619 articles were found on the prevalence and associated risk factors of IPI in Malaysia. Fifteen of these articles were excluded due to duplicates and 4436 articles were flagged as ineligible by EndNote Volume X. (Clarivate Analytics, PA, USA). Of the remaining 168 articles, 65 were excluded after reviewing the titles and abstracts of the articles. The remaining 103 articles were screened for eligibility by reading the full text. As a result, 54 articles were further excluded during data extraction, mainly because of the different outcomes of interest and because the odds ratio (OR), 95% confidence interval (CI) and number of positive cases were not reported. Therefore, 49 of the articles met the inclusion criteria and were included in this review (Table 3).

### 3.1. Characteristics and quality of studies included in the meta-analysis

Risk of bias (quality assessment) was assessed using the Joanna Briggs Institute critical appraisal tools approach was used to assess the overall quality of the evidence. This tool consists of nine criteria that assess aspects such as adequacy of sampling frame, sampling method, sample size, detail of study participants and settings, scope of data analysis, validity and reliability of condition measurement, use of appropriate statistical analysis, and adequacy of response rate. Each criterion was assessed based on the information provided in each study and rated as "Yes"," "No"," "Unclear" or "Not applicable". This quality assessment helped determine the risk of bias and overall reliability of the included studies. The quality score of the individual articles is shown in Table 3 and S6 Appendix. Of the total studies included, 32 studies (32 out of 49) received a "yes" for at least seven of the nine criteria, indicating a generally high methodological quality. The most frequently met criteria were: appropriate sample size (98% yes), appropriate sampling frame (96% yes), appropriate sampling method (94% yes) and. However, reporting was less consistent in some areas. For example, the criterion of data analysis conducted coverage rate was rated as "no" in 63% of studies due to missing or incomplete information. The statistical methods used to identify infection was also classified as "no" in 67% of the studies. This analysis emphasises the overall very good quality of the available evidence, which lends credibility to the conclusions on the prevalence and distribution of intestinal protozoa in Malaysia based on the literature reviewed.

**Table 3. Chronological overview of the 49 published population-based studies that were included in the meta-analysis review.**

| No | Author and Year | Region | Detection Method | Total sample | Overall Prevalence | Study Population | Total Score | Quality Rating |
|---|---|---|---|---|---|---|---|---|
| 1 | Siti Farah Norasyikeen et al., 2024 [28] | Kuala Lumpur | Microscopy | 134 | 0.082 | Hospitalized patients | 0 | Low risk |
| 2 | Mohd Hanapi et al., 2023 [29] | Selangor | Microscopy | 418 | 0.077 | Migrants | 0 | Low risk |
| 3 | Zamari et al., 2023 [30] | Selangor | Microscopy | 37 | 0.676 | Local community | 3 | Low risk |
| 4 | Saidin et al., 2022 [16] | Perak | Molecular | 55 | 0.164 | Aboriginal people | 0 | Low risk |
| 5 | Mohamed Kamel et al., 2022 [31] | Perak | Microscopy | 99 | 0.091 | Aboriginal people | 4 | Moderate risk |
| 6 | Tokijoh et al., 2022 [32] | Perak, | Molecular | 544 | 0.213 | Aboriginal people | 0 | Low risk |
| 7 | Tokijoh et al., 2021 [33] | Perak | Molecular | 453 | 0.647 | Aboriginal people | 0 | Low risk |
| 8 | Sahimin et al., 2020 [34] | Peninsular Malaysia | Microscopy | 206 | 0.029 | Local communities | 0 | Low risk |
| 9 | Mohamed Nur Adli & Mohamed Kamel, 2020 [35] | Pahang | Microscopy | 208 | 0.346 | Aboriginal people | 4 | Moderate risk |
| 10 | Tang et al., 2020 [36] | Perak, | Microscopy | 116 | 0.681 | Aboriginal people | 4 | Moderate risk |
| 11 | Sahimin et al., 2019 [37] | Peninsular Malaysia | Microscopy, Immunoassay, Molecular | 306 | 0.386 | Migrants | 3 | Low risk |
| 12 | Jeyaprakasam et al., 2019 [38] | Perak | Microscopy | 139 | 0.151 | Aboriginal people | 4 | Moderate risk |
| 13 | Asady Abdullah et al., 2019 [39] | Pahang | Microscopy | 135 | 0.252 | Hospitalized patients | 0 | Low risk |
| 14 | Noradilah et al., 2019 [40] | Pahang | Microscopy | 473 | 0.104 | Aboriginal people | 0 | Low risk |
| 15 | Mohamed Kamel et al., 2019 [41] | Pahang | Microscopy | 111 | 0.234 | Aboriginal people | 4 | Moderate risk |
| 16 | Hartini & Mohamed Kamel., 2018 [42] | Kuala Lumpur | Microscopy | 171 | 0.234 | Hospitalized patients | 4 | Moderate risk |
| 17 | Rajoo et al., 2017 [43] | Sarawak | Microscopy | 341 | 0.103 | Aboriginal people | 0 | Low risk |
| 18 | Saidin et al., 2017 [44] | Kelantan | Microscopy, Immunoassay, Molecular | 70 | 0.657 | Hospitalized patients | 0 | Low risk |
| 19 | Sahimin et al., 2016 [13] | Peninsular Malaysia | Microscopy | 388 | 0.255 | Migrants | 0 | Low risk |
| 20 | Elyana et al., 2016 [24] | Terengganu | Microscopy | 340 | 0.259 | Aboriginal people | 0 | Low risk |
| 21 | Chin et al., 2016 [2] | Selangor | Microscopy, Molecular | 186 | 0.172 | Aboriginal people | 0 | Low risk |
| 22 | Anuar et al., 2016 [45] | Pahang | Microscopy | 255 | 0.149 | Aboriginal people | 4 | Moderate risk |
| 23 | Mohamed Kamel et al., 2016 [46] | Kelantan | Microscopy | 111 | 0.342 | Aboriginal people | 4 | Moderate risk |
| 24 | Wong et al., 2016 [47] | Peninsular Malaysia | Immunoassay | 375 | 0.709 | Aboriginal people | 0 | Low risk |
| 25 | Nisha et al., 2015 [48] | Selangor | Microscopy | 110 | 0.073 | Aboriginal people | 4 | Moderate risk |

*(Continued)*

**Table 3.** (Continued)

| No | Author and Year | Region | Detection Method | Total sample | Overall Prevalence | Study Population | Total Score | Quality Rating |
|----|-----------------|--------|------------------|--------------|--------------------|-----------------|-------------|----------------|
| 26 | Angal et al., 2015 [22] | Selangor | Microscopy | 294 | 0.054 | Migrants | 0 | Low risk |
| 27 | Asma et al., 2015 [49] | Peninsular Malaysia | Microscopy, Molecular | 346 | 0.124 | Hospitalized patients | 0 | Low risk |
| 28 | Ahmed Al-Delaimy et al., 2014 [50] | Pahang | Microscopy | 498 | 0.476 | Aboriginal people | 0 | Low risk |
| 29 | Lee et al., 2014 [51] | Peninsular Malaysia | Microscopy | 269 | 0.182 | Aboriginal people | 0 | Low risk |
| 30 | Ahmad et al., 2014 [52] | Perak | Microscopy | 131 | 0.008 | Local communities | 5 | Moderate risk |
| 31 | Choy et al., 2014 [53] | Peninsular Malaysia | Microscopy | 1330 | 0.281 | Aboriginal people | 0 | Low risk |
| 32 | Anuar et al., 2014 [54] | Peninsular Malaysia | Microscopy, Molecular | 611 | 0.16 | Aboriginal people | 0 | Low risk |
| 33 | Hanapian et al., 2014 [55] | Perak | Microscopy | 175 | 0.594 | Aboriginal people | 4 | Moderate risk |
| 34 | Anuar et al., 2014 [56] | Peninsular Malaysia | Microscopy | 447 | 0.058 | Aboriginal people | 0 | Low risk |
| 35 | Hartini et al., 2013 [57] | Kelantan | Microscopy | 111 | 0.45 | Aboriginal people | 4 | Moderate risk |
| 36 | Lau et al., 2013 [19] | Peninsular Malaysia | Molecular | 334 | 0.195 | Aboriginal people | 0 | Low risk |
| 37 | Al-Mekhlafi et al., 2013 [58] | Pahang | Microscopy | 374 | 0.313 | Local communities | 0 | Low risk |
| 38 | Al-Harazi et al., 2013 [59] | Pahang | Microscopy | 307 | 0.365 | Aboriginal people | 4 | Moderate risk |
| 39 | Basuni et al., 2012 [60] | Kelantan | Microscopy, Molecular | 225 | 0.08 | Hospitalized patients | 0 | Low risk |
| 40 | Sinniah et al., 2012 [61] | Perak | Microscopy | 77 | 0.078 | Aboriginal people | 4 | Moderate risk |
| 41 | Ngui et al., 2012 [20] | Peninsular Malaysia | Microscopy, Molecular | 426 | 0.176 | Local communities | 0 | Low risk |
| 42 | Rossle et al., 2012 [62] | Selangor | Microscopy, Immunoassay | 130 | 0.046 | Hospitalized patients | 4 | Moderate risk |
| 43 | Anuar et al., 2012 [63] | Peninsular Malaysia | Microscopy | 500 | 0.186 | Aboriginal people | 0 | Low risk |
| 44 | Asma et al., 2011 [21] | Peninsular Malaysia | Microscopy | 346 | 0.324 | Hospitalized patients | 2 | Low risk |
| 45 | Ngui et al., 2011 [64] | Peninsular Malaysia | Microscopy | 716 | 0.228 | Local communities | 0 | Low risk |
| 46 | Lim et al., 2011 [65] | Selangor | Microscopy, Molecular | 122 | 0.279 | Hospitalized patients | 0 | Low risk |
| 47 | Lono et al., 2011 [66] | Selangor | Microscopy, Molecular | 247 | 0.04 | Hospitalized patients | 0 | Low risk |
| 48 | Al-Mekhlafi et al., 2011 [67] | Selangor | Microscopy | 276 | 0.072 | Aboriginal people | 4 | Medium risk |
| 49 | Al-Mekhlafi et al., 2010 [68] | Pahang | Microscopy | 241 | 0.216 | Aboriginal people | 4 | Medium risk |

## 3.2. Pooled intestinal protozoal infection prevalence among study participants in Malaysia

Results of the meta-analysis showed significant differences in the overall prevalence rates in different areas. The prevalence of IPI varied widely, ranging from 19% to 30% (Table 4). The lowest prevalence rate of IPI was documented by Ahmad et al. (2014) with a prevalence of 0.1% [52]. At the other end of the spectrum, the both study by Wong et al. (2016) recorded the highest prevalence rate, namely 71% [47]. The overall pooled prevalence of IPI in Malaysia was 24% (95% CI: 0.19, 0.30) (Fig 2). A high level of heterogeneity ($I^2 = 98.94\%$, $P < 0.001$) was observed, likely due to differences in study populations, geographical locations, diagnostic methods and study designs.

## 3.3. Subgroup analysis

A total of 49 studies with 14314 participants were included in this meta-analysis. Microscopy was the most commonly used detection method, reported in 35 studies with 10218 participants with a prevalence of 23% (95% CI: 0.17.0, 0.29). Molecular methods were used in 4 studies with 1386 participants, with a prevalence of 31% (95% CI: 0.08, 0.53). One study using immunoassay methods reported a prevalence of 70.9%, but this is not included in the table due to the lack of comparable data from other studies using the same standalone method. Meanwhile, 7 studies combining microscopic and molecular methods (2163 participants) reported a prevalence of 14% (95% CI: 0.09, 0.20). Two studies using a combination of microscopy, molecular and immunoassay methods (376 participants) reported a prevalence of 52% (95% CI: 0.25, 0.78).

Selangor and Kuala Lumpur had the most studies (11 studies, 2,125 participants), but reported the lowest prevalence of 16% (95% CI: 0.06, 0.25) compared to Kelantan (3 studies, 406 participants) with prevalence of 39% (95% CI: 0.06,

Table 4. Meta-analysis of the included studies.

| Group/Subgroup | Included Studies | Total sample (n) | Pooled prevalence % | 95% CI | Heterogeneity | |
|---|---|---|---|---|---|---|
| | | | | | I²% consistency | P value |
| Intestinal Protozoa Infection | 49 | 14314 | 24.0 | 0.24 [0.19, 0.30] | 98.94 | <0.001 |
| **Detection method of Intestinal Protozoa** | | | | | | |
| Microscopy (Trichrome, etc.) | 35 | 10218 | 23.0 | 0.23 [0.17, 0.29] | 98.73 | <0.001 |
| Molecular (PCR, etc.) | 4 | 1386 | 31.0 | 0.31 [0.8, 0.53] | 98.92 | <0.001 |
| Microscopy & Molecular | 7 | 2163 | 14.0 | 0.14 [0.09, 0.20] | 93.45 | <0.001 |
| Microscopy, Molecular and Immunoassay | 2 | 376 | 52.0 | 0.52 [0.25, 0.78] | 94.58 | <0.001 |
| **Year of Publication Studies** | | | | | | |
| 2020-2024 | 10 | 2770 | 30.0 | 0.30 [0.13, 0.46] | 99.32 | <0.001 |
| 2015-2019 | 17 | 4151 | 24.0 | 0.24 [0.14, 0.27] | 98.52 | <0.001 |
| 2010-2014 | 22 | 7893 | 20.0 | 0.20 [0.14, 0.27] | 98.62 | <0.001 |
| **Area of Studies** | | | | | | |
| Selangor & KL | 11 | 2125 | 16.0 | 0.16 [0.06, 0.25] | 98.65 | <0.001 |
| Perak | 9 | 1789 | 29.0 | 0.29 [0.12, 0.47] | 99.30 | <0.001 |
| Pahang | 8 | 2361 | 28.0 | 0.28 [0.21, 0.35] | 94.83 | <0.001 |
| Kelantan | 3 | 406 | 39.0 | 0.39 [0.06, 0.72] | 98.20 | <0.001 |
| Peninsular Malaysia | 14 | 6600 | 23.0 | 0.23 [0.15, 0.32] | 98.97 | <0.001 |
| **Targeted Population/ subjects** | | | | | | |
| Aboriginal people | 29 | 9092 | 27.0 | 0.27 [0.20, 0.34] | 98.76 | <0.001 |
| Migrant | 4 | 1406 | 19.0 | 0.19 [0.04, 0.34] | 98.77 | <0.001 |
| Local communities | 6 | 1890 | 23.0 | 0.23 [0.04, 0.42] | 99.58 | <0.001 |
| Patients | 10 | 1926 | 21.0 | 0.21 [0.10, 0.32] | 98.46 | <0.001 |

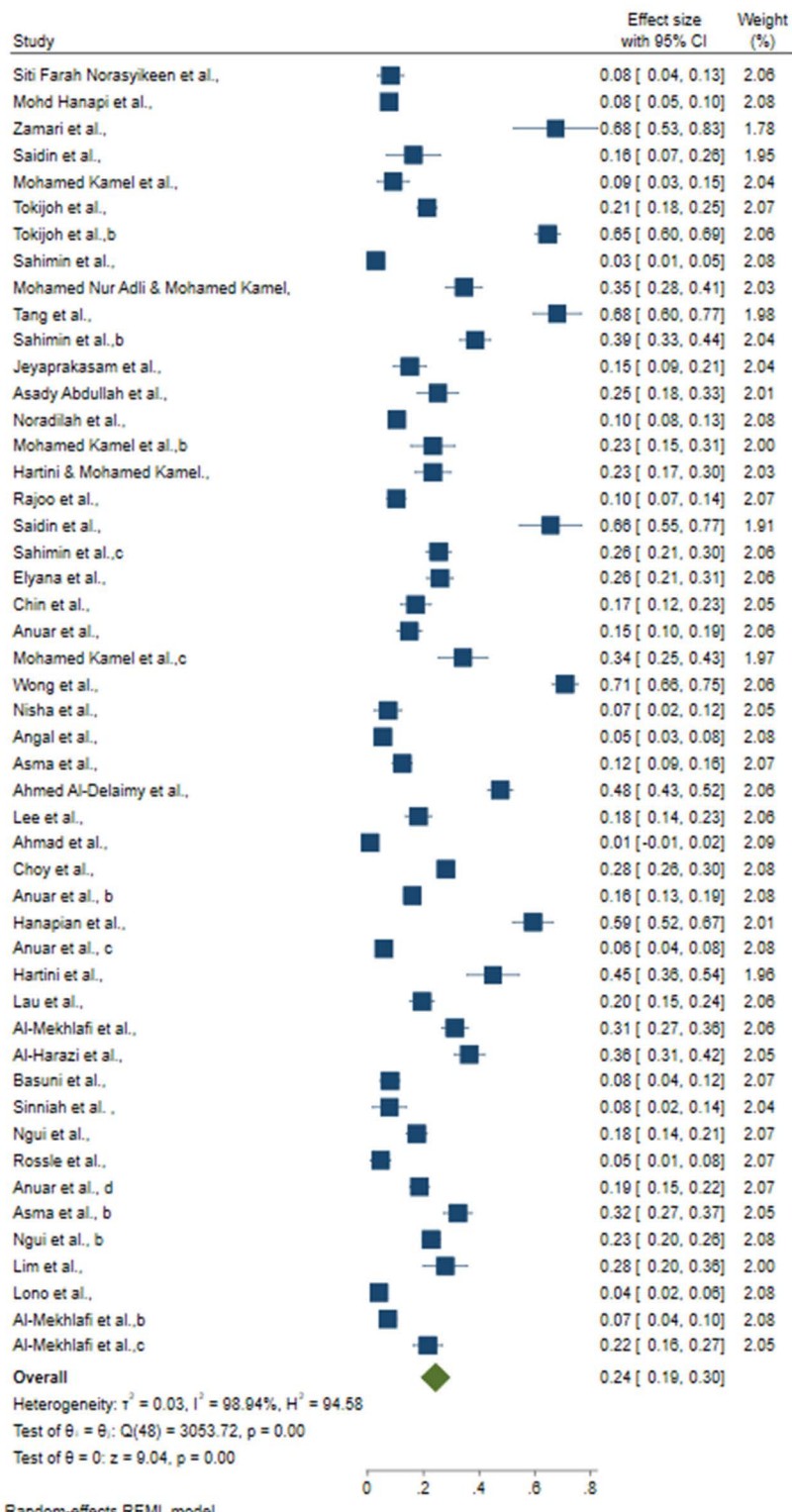

**Fig 2. Forest plot of the overall prevalence of the IPI included in 49 articles in this review.**

0.72], while Perak (9 studies, 1,789 participants) with a prevalence of 29% (95% CI: 0.12.0, 0.47), and Pahang (8 studies, 2,361 participants) showed a prevalence of 28% (95% CI: 0.21, 0.35). Peninsular Malaysia (more than one region) had 14 studies with 6600 participants reported the prevalence of 23% (95% CI: 0.15, 0.32), while East Malaysia had only two studies (341 participants) one each from Sabah and Sarawak. However, these were not included in the analysis as most of the studies conducted in East Malaysia focused on animals or environmental samples rather than human populations, making them unsuitable for inclusion in the prevalence analysis.

The prevalence of intestinal protozoa varied over time, showing a trend every five years. Studies from 2020 to 2024 (10 studies, 2770 participants) reported a prevalence of 30% (95% CI: 0.13, 0.46) Between 2015 and 2019 (17 studies, 4151 participants), the prevalence was 24% (95% CI: 0.15, 0.33). Earlier studies from 2011 to 2014 (22 studies, 7893 participants) reported a prevalence of 20% (95% CI: 0.14, 0.27). Among the different population groups, Aboriginal communities had the most studies (29 studies, 9,092 participants) with a prevalence of 27% (95% CI: 0.20, 0.34). Studies on migrant groups (4 studies, 1,406 participants) reported a prevalence of 19% (95% CI: 0.04, 0.34). Studies on local communities (6 studies, 1890 participants) showed a prevalence of 23% (95% CI: 0.04, 0.42). Studies in patient groups (10 studies, 1926 participants) showed a prevalence of 21% (95% CI: 0.10, 0.32). All forest plot diagrams for this analysis can be found in the supplementary file (S4 Appendix). Subgroup analyses by region and population revealed high heterogeneity ($I^2 > 98\%$) between the studies.

### 3.4. Common intestinal protozoal in Malaysia

Based on the analysis, the highest pooled prevalence of intestinal protozoa in Malaysia was *Entamoeba spp.* at 18% (95% CI: 0.12, 0.24) (Fig 3), followed by *G. lamblia* at 11% (95% CI: 0.08, 0.14) (Fig 4) and *Cryptosporidium spp.* at 9% (95% CI: 0.03, 0.14) (Fig 5). The analysis showed considerable heterogeneity between studies, with $I^2$ values indicating high inconsistency between studies for *Entamoeba spp.* ($I^2 = 99.24\%$), *G. lamblia* ($I^2 = 98.42\%$) and *Cryptosporidium spp.* ($I^2 = 99.75\%$) with all intestinal protozoa P-value $< 0.001$.

### 3.5. Factors associated with intestinal protozoal infection in Malaysia

Ten risk factors were analysed in a subgroup meta-analysis to identify possible predictors of protozoal intestinal infections (see Table 5). A subgroup meta-analysis revealed a higher prevalence among children under 15 years of age (47%), people from low-income households (48%) and people using untreated water sources (52%). Other notable factors were the lack of toilets (49%), poor hand hygiene (46%), contact with animals (42%) and eating raw vegetables (40%). Children under the age of 15 had the highest pooled prevalence of 47.0% (95% CI: 0.33–0.61; $I^2 = 99.64\%$), according to risk factor analysis. In terms of gender, the pooled prevalence for males was 42.0% (95% CI: 0.27–0.56; $I^2 = 99.68\%$). A significant part was played by socioeconomic factors. People with no formal education had a prevalence of 38.0% (95% CI: 0.23–0.54; $I^2 = 99.52\%$), while those from households making less than RM500 per month had a pooled prevalence of 48.0% (95% CI: 0.33–0.64; $I^2 = 99.06\%$). Risk factors related to behaviour and the environment were also important. A pooled prevalence of 52.0% (95% CI: 0.33–0.70; $I^2 = 99.74\%$) was linked to the use of untreated water sources, while a prevalence of 49.0% (95% CI: 0.31–0.67; $I^2 = 99.74\%$) was linked to the absence of toilet availability. The pooled prevalence among those who reported eating raw vegetables was 40.0% (95% CI: 0.18–0.62; $I^2 = 98.64\%$). The significance of personal hygiene was further supported by the 46.0% prevalence of people who did not wash their hands (95% CI: 0.25–0.66; $I^2 = 99.77\%$). A pooled prevalence of 42.0% (95% CI: 0.22–0.62; $I^2 = 99.80\%$) was linked to contact with domestic or stray animals. High heterogeneity ($I^2 > 98\%$), indicating differences between the studies and the influence of unmeasured moderators. The results should be interpreted with caution as the analysis focused on the most frequently reported risk factors, while other unmeasured influences may still have affected the results.

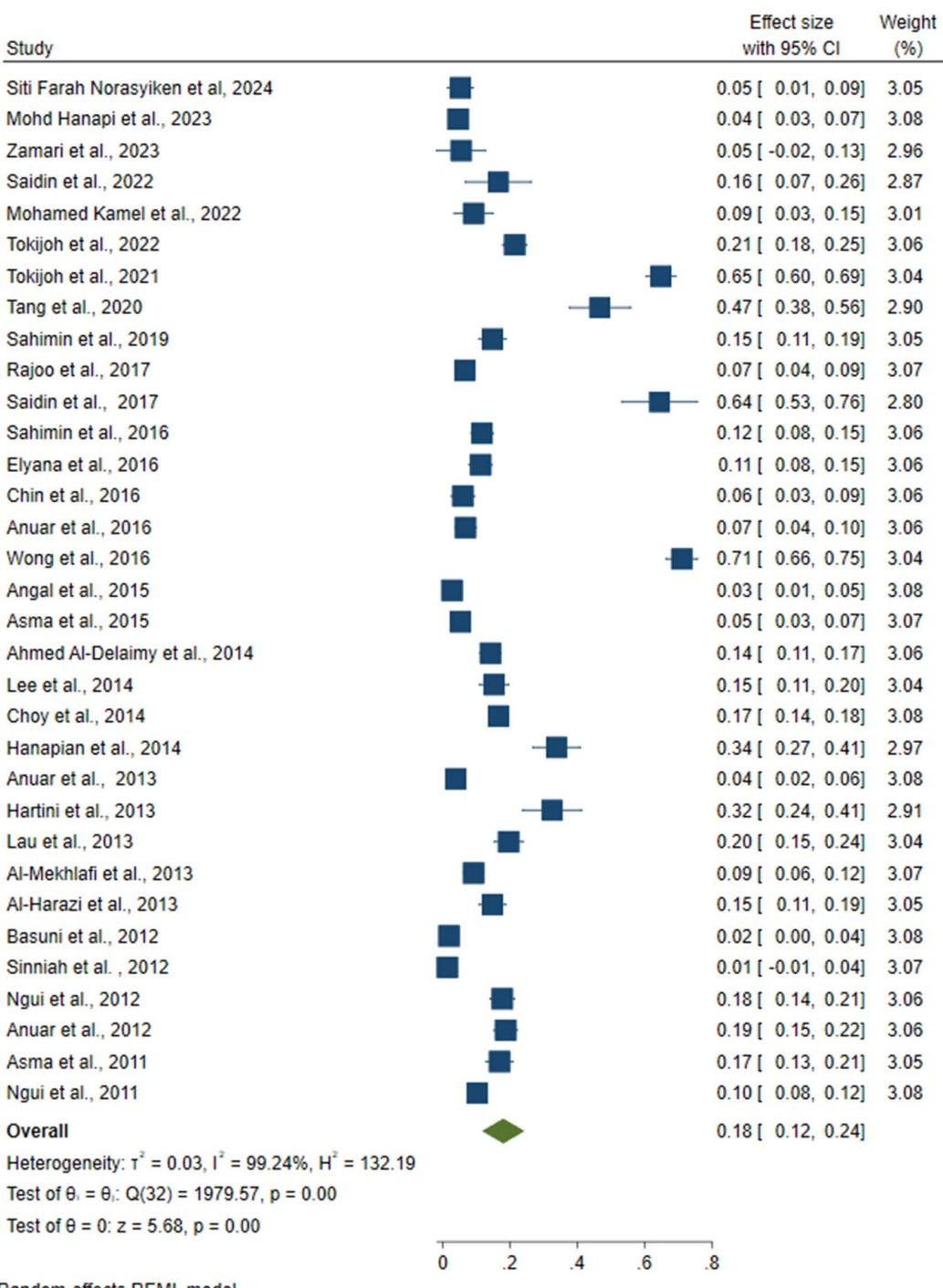

**Fig 3. The pooled prevalence of the *Entamoeba spp*. reported in Malaysia.**

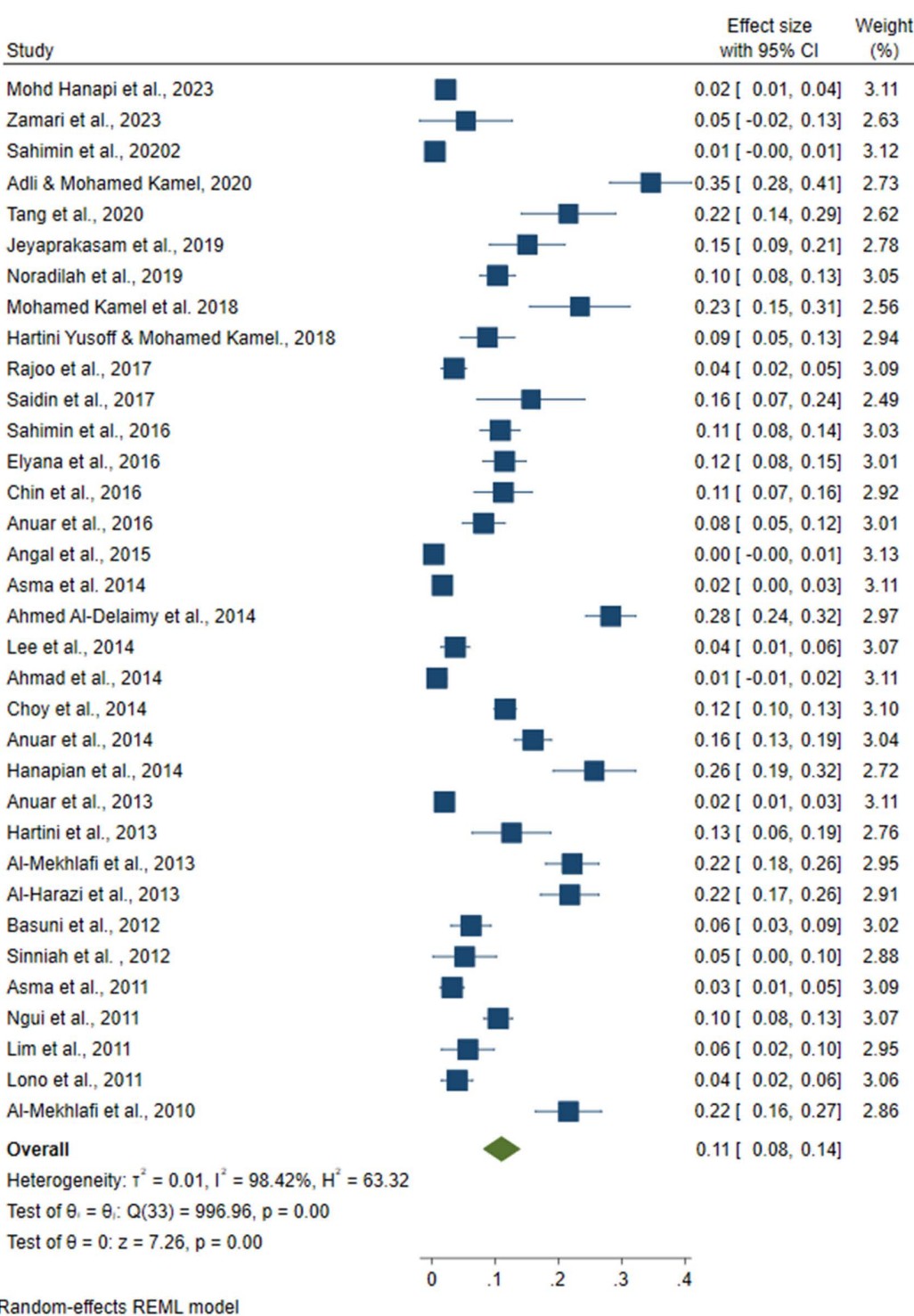

| Study | Effect size with 95% CI | Weight (%) |
|---|---|---|
| Mohd Hanapi et al., 2023 | 0.02 [ 0.01, 0.04] | 3.11 |
| Zamari et al., 2023 | 0.05 [ -0.02, 0.13] | 2.63 |
| Sahimin et al., 20202 | 0.01 [ -0.00, 0.01] | 3.12 |
| Adli & Mohamed Kamel, 2020 | 0.35 [ 0.28, 0.41] | 2.73 |
| Tang et al., 2020 | 0.22 [ 0.14, 0.29] | 2.62 |
| Jeyaprakasam et al., 2019 | 0.15 [ 0.09, 0.21] | 2.78 |
| Noradilah et al., 2019 | 0.10 [ 0.08, 0.13] | 3.05 |
| Mohamed Kamel et al. 2018 | 0.23 [ 0.15, 0.31] | 2.56 |
| Hartini Yusoff & Mohamed Kamel., 2018 | 0.09 [ 0.05, 0.13] | 2.94 |
| Rajoo et al., 2017 | 0.04 [ 0.02, 0.05] | 3.09 |
| Saidin et al., 2017 | 0.16 [ 0.07, 0.24] | 2.49 |
| Sahimin et al., 2016 | 0.11 [ 0.08, 0.14] | 3.03 |
| Elyana et al., 2016 | 0.12 [ 0.08, 0.15] | 3.01 |
| Chin et al., 2016 | 0.11 [ 0.07, 0.16] | 2.92 |
| Anuar et al., 2016 | 0.08 [ 0.05, 0.12] | 3.01 |
| Angal et al., 2015 | 0.00 [ -0.00, 0.01] | 3.13 |
| Asma et al. 2014 | 0.02 [ 0.00, 0.03] | 3.11 |
| Ahmed Al-Delaimy et al., 2014 | 0.28 [ 0.24, 0.32] | 2.97 |
| Lee et al., 2014 | 0.04 [ 0.01, 0.06] | 3.07 |
| Ahmad et al., 2014 | 0.01 [ -0.01, 0.02] | 3.11 |
| Choy et al., 2014 | 0.12 [ 0.10, 0.13] | 3.10 |
| Anuar et al., 2014 | 0.16 [ 0.13, 0.19] | 3.04 |
| Hanapian et al., 2014 | 0.26 [ 0.19, 0.32] | 2.72 |
| Anuar et al., 2013 | 0.02 [ 0.01, 0.03] | 3.11 |
| Hartini et al., 2013 | 0.13 [ 0.06, 0.19] | 2.76 |
| Al-Mekhlafi et al., 2013 | 0.22 [ 0.18, 0.26] | 2.95 |
| Al-Harazi et al., 2013 | 0.22 [ 0.17, 0.26] | 2.91 |
| Basuni et al., 2012 | 0.06 [ 0.03, 0.09] | 3.02 |
| Sinniah et al. , 2012 | 0.05 [ 0.00, 0.10] | 2.88 |
| Asma et al., 2011 | 0.03 [ 0.01, 0.05] | 3.09 |
| Ngui et al., 2011 | 0.10 [ 0.08, 0.13] | 3.07 |
| Lim et al., 2011 | 0.06 [ 0.02, 0.10] | 2.95 |
| Lono et al., 2011 | 0.04 [ 0.02, 0.06] | 3.06 |
| Al-Mekhlafi et al., 2010 | 0.22 [ 0.16, 0.27] | 2.86 |
| **Overall** | 0.11 [ 0.08, 0.14] | |

Heterogeneity: $\tau^2 = 0.01$, $I^2 = 98.42\%$, $H^2 = 63.32$

Test of $\theta_i = \theta_j$: Q(33) = 996.96, p = 0.00

Test of $\theta = 0$: z = 7.26, p = 0.00

Random-effects REML model

**Fig 4. The pooled prevalence of the *Giardia spp*. reported in Malaysia.**

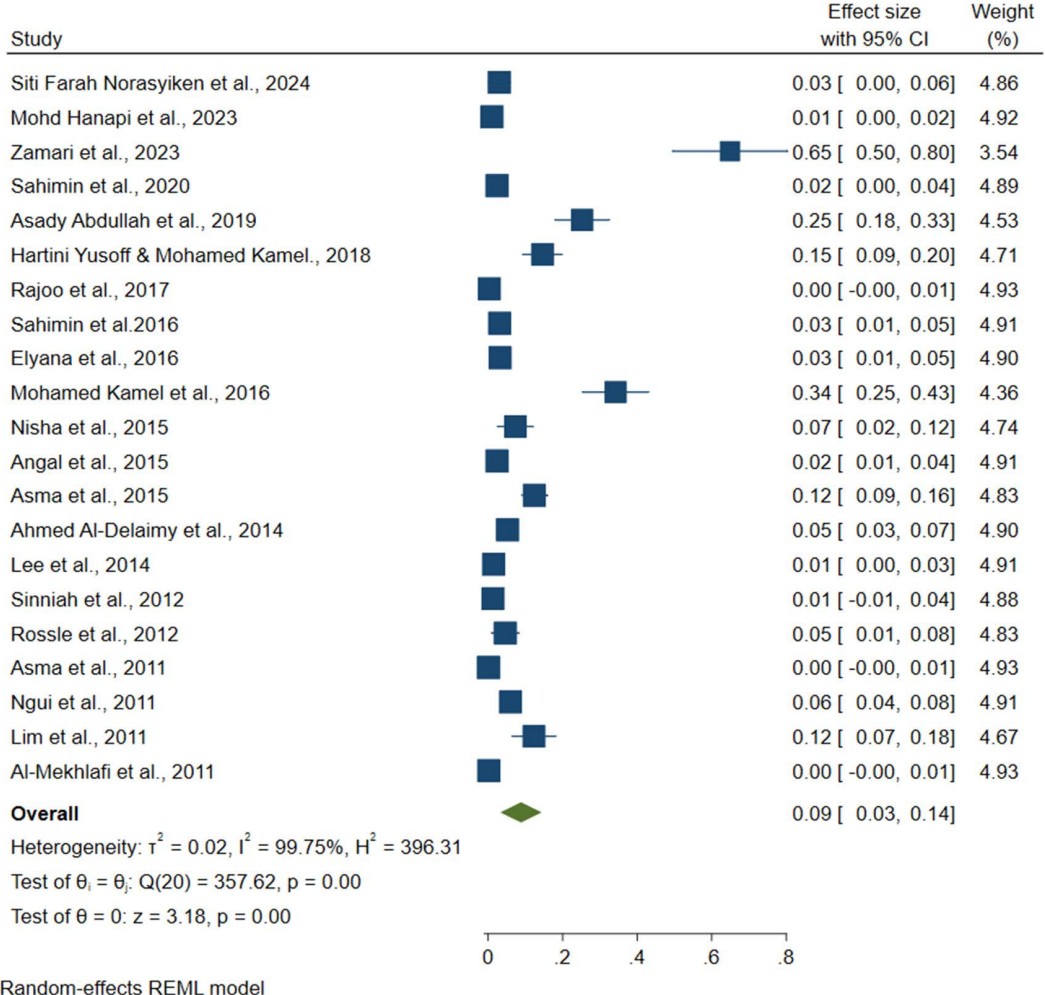

Fig 5. **The pooled prevalence of the *Cryptosporidium spp*. reported in Malaysia.**

### 3.6. Sensitivity analysis

A sensitivity analysis was carried out using the criteria for excluding outliers. This involved re-analysing the data with and without the outlier studies to determine how they affected the pooled prevalence estimates, confidence intervals (95% CI) and heterogeneity ($I^2$). After eliminating the outliers, the analysis showed an overall improvement in heterogeneity ($I^2$) in all subgroups, indicating higher precision and reliability of the pooled estimates (Table 6 and S7 Appendix). For example, overall prevalence decreased from 24% ($I^2 = 98.94\%$) to 18% ($I^2 = 96.67\%$), microscopic detection from 23% to 16% and molecular detection from 31% to 20%, the latter showing a sharp decrease in $I^2$ from 98.92% to 0.06%. The analysis by year also showed a significant decrease in heterogeneity ($I^2 = 99.32\%$ to 71.94%) and a sharp decrease in pooled prevalence for 2020–2024 (from 30% to 7%). Geographic subgroups such as Selangor and Kuala Lumpur showed similar patterns, with prevalence decreasing from 16% to 6% and $I^2$ from 98.65% to 21.42%. The direction and general pattern of the results were consistent with the original analysis, although there was a marked decrease in heterogeneity and minor changes in prevalence in some groups. For example, even after removing outliers, the prevalence of infection remained higher in population groups such as children, people on low incomes and people relying on untreated water. Although

**Table 5. Meta-analysis of the risk factors associated with IPI in Malaysia.**

| Group/Subgroup | Pooled prevalence % | 95% CI | Heterogeneity | | Interpretation |
|---|---|---|---|---|---|
| | | | I²% con-sistency | P value | |
| **Age (<15 years old)** | 47.0 | 0.47 [0.33, 0.61] | 99.64 | <0.001 | Children have a notably higher prevalence, suggesting they are a vulnerable group likely due to immature hygiene practices and exposure. |
| **Gender (Male)** | 42.0 | 0.42 [0.27, 0.56] | 99.68 | <0.001 | Males show a higher prevalence, possibly linked to behavioral or occupational exposure differences. |
| **Household income (<RM500)** | 48.0 | 0.48 [0.33, 0.64] | 99.06 | <0.001 | Individuals from low-income households are at increased risk, highlighting socioeconomic disparities in infection burden. |
| **Education status (no formal education)** | 38.0 | 0.38 [0.23, 0.54] | 99.52 | <0.001 | Lack of formal education is associated with higher infection prevalence, possibly due to limited awareness of hygiene and sanitation. |
| **Water sources (untreated)** | 52.0 | 0.52 [0.33, 0.70] | 99.74 | <0.001 | More than half of individuals using untreated water are infected, underlining the importance of safe water access. |
| **Sanitation facility (No toilet availability)** | 49.0 | 0.49 [0.31, 0.67] | 99.74 | <0.001 | Poor sanitation is strongly associated with increased infection rates, indicating the role of environmental hygiene. |
| **Eating raw vegetables (Yes)** | 40.0 | 0.40 [0.18, 0.62] | 98.64 | <0.001 | Consuming raw vegetables may contribute to infection, likely due to contamination with protozoa cysts. |
| **No Handwashing practice (Yes)** | 46.0 | 0.46 [0.25, 0.66] | 99.77 | <0.001 | Poor hand hygiene significantly raises the risk of infection, rein-forcing the need for hygiene education. |
| **Contact with domestic or stray animals (Yes)** | 42.0 | 0.42 [0.22, 0.62] | 99.80 | <0.001 | Animal contact is a likely transmission route, possibly due to zoonotic protozoa or contaminated environments. |

prevalence rates were slightly reduced (e.g., from 27% to 18% in the indigenous population), target groups such as hospitalised patients and the indigenous population remained at risk. These results suggest that although the elimination of outlier studies increased statistical precision and reduced heterogeneity, the primary conclusions of the meta-analysis remained largely unchanged.

### 3.7. Meta-regression model

The impact of various study-level covariates on the pooled prevalence estimates was examined using meta-regression analyses (S8 Appendix). With p-values of 0.305 for microscopic and 0.285 for molecular methods, the detection method did not significantly correlate with prevalence. Selangor (p = 0.011) and Peninsular Malaysia (p = 0.043) were found to have significantly lower prevalences than the reference region, but the overall model was not statistically significant for geo-graphical region (p = 0.205). Hospitalised patients (p = 0.465), local communities (p = 0.739), and migrants (p = 0.871) did not differ significantly from one another. Similarly, p-values of 0.223 for the 2020–2024 group and 0.717 for the 2015–2019 group indicated no significant effects for study year. The JBI score, which measures study quality, also revealed no signif-icant effect (p = 0.937), and the prevalence did not significantly differ between age groups >15 and ≤15 years (p = 0.940). With p-values of 0.148 for *E. histolytica* and 0.436 for *G. lamblia*, analysis by type of protozoa detected revealed no significant associations. All models maintained a high level of residual heterogeneity, with $R^2$ values ranging from 0.00% to 4.59% and $I^2$ values exceeding 98%. The summarised data shown Table 7.

### 3.8. Risk of publication bias included in the meta-analysis

An analysis of the funnel plot (Fig 6) indicates a possible publication bias, which is evident in its asymmetry and the clustering of smaller studies around an effect size of approximately 0.2 with standard errors between 0.01 and 0.06. This visual pattern suggests that smaller studies with significant or positive results may be overrepresented. To confirm

**Table 6. Sensitivity analysis of pooled prevalence from various subgroup by excluding the studies outlier.**

| Subgroup | After outlier | | | Comparison with original data | | |
|---|---|---|---|---|---|---|
| | Studies (n) | IPI [95% CI] | I² (%) | Studies (n) | IPI [95% CI] | I² (%) |
| **Overall study** | 27 | 0.18 [0.14, 0.21] | 96.67 | 49 | 0.24 [0.19, 0.30] | 98.94 |
| **Detection method of Intestinal Protozoa** | | | | | | |
| Microscopy | 27 | 0.16 [0.12, 0.19] | 96.48 | 35 | 0.23 [0.17, 0.29] | 98.73 |
| Molecular | 3 | 0.20 [0.18, 0.23] | 0.06 | 4 | 0.31 [0.8, 0.53] | 98.92 |
| Microscopy and Molecular | 4 | 0.16 [0.13, 0.18] | 40.62 | 7 | 0.14 [0.09, 0.20] | 93.45 |
| **Year of Publication Studies** | | | | | | |
| 2020-2024 | 5 | 0.07 [0.04, 0.11] | 71.94 | 10 | 0.30 [0.13, 0.46] | 99.32 |
| 2015-2019 | 13 | 0.16 [0.12, 0.20] | 91.36 | 17 | 0.24 [0.14, 0.27] | 98.52 |
| 2010-2014 | 15 | 0.21 [0.16, 0.25] | 95.30 | 22 | 0.20 [0.14, 0.27] | 98.62 |
| **Area of Studies** | | | | | | |
| Selangor and KL | 7 | 0.06 [0.05, 0.07] | 21.42 | 11 | 0.16 [0.06, 0.25] | 98.65 |
| Perak | 9 | 0.14 [0.09, 0.19] | 77.19 | 9 | 0.29 [0.12, 0.47] | 99.30 |
| Pahang | 7 | 0.30 [0.25, 0.34] | 72.94 | 8 | 0.28 [0.21, 0.35] | 94.83 |
| Kelantan | 2 | 0.55 [0.35, 0.75] | 87.32 | 3 | 0.39 [0.06, 0.72] | 98.20 |
| Peninsular Malaysia | 11 | 0.23 [0.18, 0.27] | 94.16 | 14 | 0.23 [0.15, 0.32] | 98.97 |
| **Targeted Population/ subjects** | | | | | | |
| Aboriginal people | 15 | 0.18 [0.16, 0.21] | 85.85 | 29 | 0.27 [0.20, 0.34] | 98.76 |
| Migrants | 2 | 0.07 [0.04, 0.09] | 34.74 | 4 | 0.19 [0.04, 0.34] | 98.77 |
| Local communities | 3 | 0.24 [0.16, 0.31] | 92.15 | 6 | 0.23 [0.04, 0.42] | 99.58 |
| Hospitalized patients | 7 | 0.19 [0.11, 0.27] | 95.14 | 10 | 0.21 [0.10, 0.32] | 98.46 |

**Table 7. Multivariable meta-regression model of IPI estimates from pooled prevalence rate cases in Malaysia.**

| Moderator | Category | Coefficient (β) | 95% CI | p-value | R² (%) | Residual I² (%) |
|---|---|---|---|---|---|---|
| **Detection Method** | Microscopy vs *Ref* | +0.1831 | (−0.167, 0.533) | 0.305 | 0.00 | 98.80 |
| | Molecular vs *Ref* | +0.1955 | (−0.163, 0.553) | 0.285 | | |
| **Region** | Selangor vs *Ref* | −0.2202 | (−0.389, −0.051) | 0.011 | 4.59 | 98.66 |
| | Peninsular Malaysia vs *Ref* | −0.1672 | (−0.329, −0.005) | 0.043 | | |
| **Population Group** | Migrant vs *Ref* | −0.0141 | (−0.184, 0.156) | 0.871 | 0.00 | 98.77 |
| | Local community vs *Ref* | −0.0253 | (−0.174, 0.123) | 0.739 | | |
| | Hospitalized vs *Ref* | −0.0408 | (−0.150, 0.069) | 0.465 | | |
| **Year Group** | 2015–2019 vs *Ref* | +0.0226 | (−0.100, 0.145) | 0.717 | 0.00 | 98.93 |
| | 2020–2024 vs *Ref* | +0.0857 | (−0.052, 0.224) | 0.223 | | |
| **Study Quality** | Moderate vs Low Risk | −0.0045 | (−0.117, 0.109) | 0.937 | 0.00 | 98.93 |
| **Age Group** | >15 years vs ≤ 15 years | +0.013 | (−0.325, 0.351) | 0.940 | 0.00 | 99.59 |
| **Protozoa Type** | EH vs *Ref* | +0.0755 | (−0.027, 0.178) | 0.148 | 0.19 | 98.87 |
| | GL vs *Ref* | +0.0409 | (−0.062, 0.144) | 0.436 | | |

β = regression coefficient; CI = confidence interval; R² = proportion of variance explained; I² = residual heterogeneity. Detection method (Ref = immunoassay); Region (Ref = Kelantan); Population group (Ref = Aboriginal population); Year group (Ref = ≤2014); Study quality (Ref = Low risk); Protozoa type (Ref = Cryptosporidium spp.).

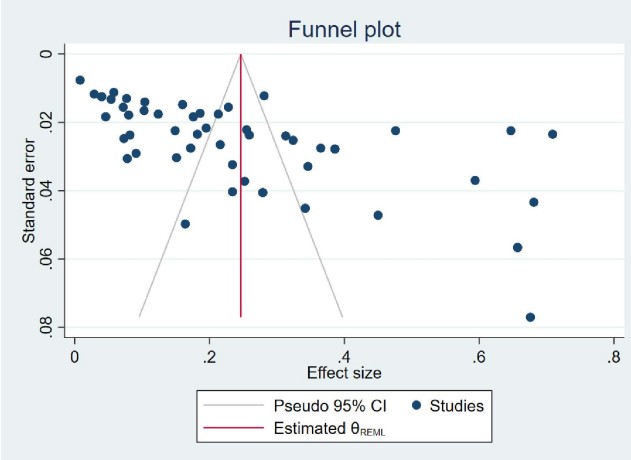

**Fig 6. Trim-and-fill analysis estimating the number of possible missing studies for risk of publication.**

this statistically, the Egger test's method for detecting effects of small studies was performed and yielded a significant result (P ≤ 0.001), indicating the presence of publication bias. The significant result of the Egger test is consistent with the visual interpretation of the funnel plot, reinforcing the evidence of publication bias in this meta-analysis. Nevertheless, the sensitivity analysis showed that the pooled prevalence estimates remained stable when individual studies were excluded, indicating the robustness of the overall results.

## 4. Discussion

This review of IPI in Malaysia provides important insights into the epidemiology and associated risk factors for these infections in different areas and population groups in Malaysia. The overall prevalence of IPI in Malaysia is 24% (95% CI: 0.19, 0.30) suggesting a high infection or burden of these infections in Malaysia. This percentage was higher than in the studies conducted in Thailand (13.9%) [69]. However, it was lower than in the studies from the Philippines (58.7%), Laos (75%), Cambodia (55.2%) and Vietnam (34.5%) [70,71]. It is noteworthy that the prevalence rates vary widely across locations: 0.8% to 98.4%. This suggests that the significant regional differences are likely to be influenced by various risk factors such as environmental conditions, hygiene practices, access to clean water and socioeconomic factors. Previous studies on prevalence in Malaysia are estimated to be significant, with the overall prevalence of protozoal infections in rural communities in Peninsular Malaysia ranging from approximately 21.4% to 30.1% [29,43,64]. In particular, *G. lamblia* is the predominant protozoan infection with prevalence rates ranging from 10.4% to 14.2%, while *E. histolytica/dispar* follows closely with rates of approximately 10.2%. *Cryptosporidium spp.* has a prevalence of approximately 2.1% in Malaysia [34]. These infections remain a major public health concern, especially in impoverished and underprivileged communities with a high overall prevalence of intestinal parasitic infections (73.2%) in both rural and remote western Malaysia.

Various diagnostic methods were used across the studies, with microscopy being the most common in Malaysia due to its low cost and simplicity [72]. Although still considered the gold standard [73,74], microscopy has limited sensitivity in low-intensity infections. Immunoassays like ELISA are available for *Giardia* and *Cryptosporidium*, but their use is limited as they cannot differentiate closely related species such as *E. histolytica* and *E. dispar* [75,76]. PCR-based molecular methods offer higher sensitivity and specificity and are increasingly combined with microscopy to improve diagnostic accuracy [74,77,78]. This trend is reflected in our findings where studies using only molecular technique reported a higher pooled prevalence of 31% (n = 4, 1386 samples) compared with 23.0% in studies using only microscopy methods (n = 10, 218 samples). The over-reliance on microscopy in more than half of the included studies likely contributed to this

inflated prevalence. Although microscopy is affordable and widely available, it is not as sensitive and specific as molecular methods and cannot distinguish *E. histolytica* from non-pathogenic *E. dispar*, possibly leading to an overestimation of prevalence. Its limitations in detecting low-intensity or mixed infections may also affect the accuracy of species-specific prevalence estimates. Interestingly, the combination of microscopy and molecular methods yielded a lower prevalence (14%; n = 7, 2163 samples), possibly due to differences in study design, population characteristics, or more stringent diagnostic criteria. Notably, the highest prevalence (52%; n = 2, 376 samples) was found in studies that employed a combination of microscopy, molecular, and immunoassay methods. This suggests that integrating multiple diagnostic techniques enhances detection capability by improving both sensitivity and specificity. Although the heterogeneity in this group was high ($I^2$ = 94.58%), the consistent trend toward higher prevalence underscores the value of multi-method diagnostic approaches in providing a more accurate estimate of the true burden of intestinal protozoal infections.

The prevalence of IPI has shown a steady increase in three consecutive periods: 20% (95% CI: 0.14, 0.27) from 2010 to 2015, 24% (95% CI: 0.14, 0.27) from 2015 to 2019 and 30% (95% CI: 0.13, 0.46) from 2020 to 2024. This upward trend can be attributed to advances in diagnostic techniques, particularly the increasing use of molecular methods such as multiplex and real-time PCR, which became more accessible and widely used during the COVID-19 pandemic [79–81]. The integration of molecular tools alongside traditional microscopy has improved the sensitivity and specificity of IPI detection, as demonstrated by studies utilising combined diagnostic approaches, including microscopy, immunoassays and PCR-based methods [37,38,44,78]. In contrast, the prevalence of IPI was lower between 2011 and 2015 (20%), probably due to a predominant reliance on microscopy, which requires skilled personnel for accurate diagnosis. Limited public awareness, poor sanitation in rural areas, socioeconomic inequalities and other risk factors may also have contributed to underreporting and lower detection rates during this period [23].

This trend differs from studies conducted in certain regions of Malaysia. In the Selangor and Kuala Lumpur regions, the pooled prevalence was 16% (95% CI: 0.06, 0.25) lower than other states, which may be due to better urban infrastructure, better access to healthcare and greater awareness of infection prevention [34]. In contrast, the prevalence in Perak was 29% (95% CI: 0.12, 0.47) and Pahang was 28% (95% CI: 0.21, 0.35) possibly due to a combination of effective public health interventions and different environmental conditions affecting parasite transmission [82]. In contrast, prevalence was higher in Kelantan at 39% (95% CI: 0.06, 0.72) likely due to the presence of rural and remote communities with limited access to healthcare and sanitation [82]. The overall prevalence in Peninsular Malaysia was 23% (95% CI: 0.15, 0.32) significantly higher than in East Malaysia. The higher prevalence in Peninsular Malaysia can be attributed to the higher population density and greater mobility of people, which facilitates transmission. However, it is important to acknowledge that Peninsular Malaysia is over-represented in this review (14 studies), whereas only two study was available from East Malaysia. Most studies located in East Malaysia mainly focused on animals or environmental samples rather than human populations, making them unsuitable for inclusion in the prevalence analysis. This imbalance may limit the generalisability of the findings and could underestimate the true burden of intestinal protozoal infections in East Malaysia. The lack of data from East Malaysia also highlights the need for more regionally representative studies to better inform nationwide public health strategies [45,53].

The different prevalence in the individual regions can be explained by differences in urbanization, infrastructure, access to healthcare and sanitation. Selangor and Kuala Lumpur, which are more urbanized, benefit from better public health infrastructure, greater hygiene awareness and easier access to healthcare, which contributes to lower prevalence rates [83]. In contrast, Perak, Pahang and Kelantan have more rural or remote areas where limited access to clean water, sanitation and medical care can increase the risk of infection [16,35,44]. In addition, environmental factors and local public health measures can influence transmission dynamics and contribute to the observed regional differences. To improve IPI surveillance, it is essential to implement standardized diagnostic protocols, especially incorporating molecular techniques such as PCR, which provide greater accuracy in species identification and detection of low-intensity or mixed infections [84,85]. The integration of molecular diagnostics with microscopy or immunoassays can improve both sensitivity and

specificity, leading to more reliable national surveillance data. In addition, establishing national guidelines for IPI diagnosis and strengthening laboratory capacity, particularly in underserved regions, will support more consistent and comparable reporting across studies.

The prevalence of IPI varies between different population groups, with aboriginal people having the highest prevalence at 27% (95% CI: 0.20, 0.34) can be attributed to inadequate sanitation, poor personal hygiene and limited health education, all of which contribute to persistent transmission [51,53,86]. The prevalence was also very high in local communities, the majority of whom are from rural areas, at 23% (95% CI: 0.04, 0.42), and hospitalised patients at 21% (95% CI: 0.10, 0.32) which is which is probably due to limited access to clean water and sanitation facilities in rural settings, and the weakened immune status or underlying health conditions among hospitalised individuals that may increase susceptibility to infection [12,29]. In contrast, prevalence in migrants' population in Malaysia, was lower at 19% (95% CI: 0.04, 0.34), reflecting possible better health screening practices upon entry, temporary living conditions with improved sanitation, or underrepresentation in studies due to accessibility and documentation barriers [24,58,64,87].

In Malaysia, *Entamoeba spp*. had the highest prevalence of intestinal protozoal infections (18%), followed by *Giardia lamblia* (11%) and *Cryptosporidium spp.* (9%). These results are consistent with previous studies, such as those conducted in Pahang, Malaysia, which reported a similar prevalence of *Entamoeba spp*. (18.5%) [88]. The observed differences in prevalence across regions may be influenced by research focus and data availability. Of the 49 studies analysed, 33 reported prevalence data for *Entamoeba spp.* suggesting that this protozoan is more prevalent compared to others. This greater research focus likely contributes to the more reliable prevalence estimates for *Entamoeba spp.* In contrast, *Cryptosporidium spp*. had the lowest reported prevalence and remains under-diagnosed and under-reported. This is partly because the detection of *Cryptosporidium spp.* requires specialised diagnostic techniques, such as modified Ziehl-Neelsen staining or molecular assays, which may not be as widely used as methods for the detection of *Entamoeba spp.* and *G. lamblia* [39]. Of the 49 studies analysed, only 21 reported prevalence data for *Cryptosporidium* spp, further highlighting the limited data availability for this protozoan.

To find possible predictors of intestinal protozoal infection (IPI) in Malaysia, we conducted a subgroup meta-analysis of ten risk factors in this review. Almost one in four people are infected as shown by the pooled prevalence of 24.0% in the analysis. The results of all subgroups consistently showed high heterogeneity ($I^2 > 98$), This heterogeneity likely reflects differences in diagnostic methods, study conditions, population characteristics, and unmeasured confounders, indicating that while the pooled prevalence provides a summary estimate, it should be interpreted cautiously in light of these variations. Children under the age of 15 had the highest pooled prevalence of 47.0%, making age a determining factor among all risk factors. Their still-developing immune systems, poor hygiene habits and increased exposure to contaminated environments especially in places with inadequate sanitation are likely causes of this finding [89]. There were also gender differences, with men having a higher prevalence (42.0%), which may be related to a higher risk of exposure in the workplace or environment. The risk of infection was significantly influenced by socioeconomic factors. The pooled prevalence among those with no formal education was 38.0%, while the prevalence among those with a household income of less than RM500 per month was even higher at 48.0%. These associations illustrate how low levels of education and poverty can lead to a lack of awareness of proper hygiene and limited access to sanitation and clean water. These conclusions arise from consistent patterns observed in the included studies, which show that individuals in rural areas with limited access to clean water and sanitation and with lower awareness of personal hygiene are at higher risk of IPI. Taken together, these findings emphasize the importance of addressing both socioeconomic inequalities and hygiene practices in the control of protozoal intestinal infections in Malaysia.

The risk of infection was also strongly influenced by behaviour and environmental factors. The critical role of water, sanitation and hygiene (WASH) infrastructure was emphasised by the highest prevalence (52.0%) associated with the use of untreated water, closely followed by the lack of toilets (49.0%) [90]. Consumption of raw vegetables (40.0%) and lack of hand washing (46.0%) were other behavioural risks that increase the risk of protozoan cyst infection. Finally, a prevalence

of 42.0% was associated with contact with domestic or stray animals, suggesting a zoonotic or indirect route of transmission through the environment. All of these results point to the multifaceted nature of IPI risk and the need for integrated interventions focussed on reducing poverty, improving sanitation, educating people about good hygiene and providing access to safe water for high-risk groups.

Giardia infections are closely linked to poor hygiene and sanitation, especially through faecal contamination of drinking water and food [91]. Inadequate water treatment and improper waste disposal favour the transmission of the parasites, with a significant proportion of cases (33.9%) attributable to the consumption of untreated water. Household size also plays a role, with 57.4 of infections occurring in people living in low-income households (<RM500), suggesting that overcrowding and limited sanitation facilities contribute to the risk. In addition, eating raw or unwashed vegetables and fruits can expose people to Giardia cysts on contaminated surfaces [92]. Direct transmission from animals is less common, but poor hygiene practices after contact with animals can increase susceptibility.

Cryptosporidium infection is influenced by several factors, including malnutrition, geographical location and water quality. Individuals from low-income households (<RM500) are at higher risk as financial constraints can limit access to clean water and adequate sanitation. In addition, 40.6% of infections were associated with a lack of toilets, which increases the likelihood of environmental contamination. Drinking untreated or contaminated water remains an important risk factor, as Cryptosporidium can survive in water sources that are not properly filtered or disinfected [83]. The association between low levels of education (45.3% of cases) and Cryptosporidium infection suggests that awareness of proper hygiene and prevention measures may also play a crucial role in transmission.

By eliminating outlier studies, we performed a sensitivity analysis to determine how they affected the results. We found that the pooled prevalence became more accurate and the overall consistency between studies improved (lower $I^2$ values) after these outliers were eliminated. For example, heterogeneity ($I^2$) improved from 98.94% to 96.67%, while overall prevalence decreased from 24% to 18%. In the molecular detection group, prevalence fell from 31% to 20% and heterogeneity from 98.92% to almost 0%, suggesting far more reliable results. The most recent studies (2020–2024) showed the same pattern, with $I^2$ improving from 99.32% to 71.94% and prevalence falling from 30% to 7%. Subgroup analyses by population and location also showed comparable improvements. For example, prevalence in the local population fell from 27% to 18% and from 16% to 6% in Selangor and Kuala Lumpur. Despite these changes, the main conclusions remained the same: high infection rates were still observed in low-income households, among children and among people who use untreated water. This suggests that while the elimination of outliers has improved the reliability of the data, the overall results of the study remain unchanged.

A meta-regression analysis was used to assess the impact of various study-level factors on the reported prevalence of intestinal protozoal infection (IPI) in Malaysia. Covariates included the type of protozoa detected, age group, study year, study quality, population group, detection method and geographical location. Only geographic location had a significant effect on these variables; studies in Peninsular Malaysia and Selangor showed a significantly lower prevalence than in the reference region. Detection method, population type (e.g., hospitalised patients, local communities, migrants), study year, study quality, age group and protozoan species showed no significant associations. All models showed a high degree of residual heterogeneity, suggesting that these factors were not sufficient to explain the differences in prevalence estimates. These results emphasise the influence of regional variables on infection rates and the need to account for geographical variance in subsequent studies of IPI in Malaysia.

The Egger test yielded a statistically significant result (P ≤ 0.001), indicating the presence of effects of smaller studies and suggesting a possible publication bias in the meta-analysis. This was supported by the asymmetry observed in the funnel plot (Fig 6), which showed an overrepresentation of smaller studies with higher prevalence rates. To assess the impact of this bias, a sensitivity analysis was performed by systematically excluding individual studies and recalculating the pooled estimates. The minimal deviation observed in these recalculations indicates that no single study unduly influenced the overall results. This strengthens the reliability of the findings, suggesting that the pooled prevalence rates are stable and

not driven by outliers or highly influential studies. However, the presence of asymmetry in the funnel plot and the significant Egger test highlight that publication bias cannot be completely excluded. Therefore, while the potential for publication bias exists, the consistency observed in the sensitivity analysis supports the credibility of the meta-analysis conclusions.

In terms of clinical and public health practise, our findings emphasise the need for measures that go beyond general improvements in sanitation and hygiene. An important step is the implementation of routine stool screening in indigenous settlements and rural schools, using rapid diagnostic tools to enable early detection and limit transmission. Strengthening diagnostic capacity is also crucial. The introduction of affordable molecular-based methods in district hospitals and teaching laboratories would improve accuracy, as many existing studies still rely on microscopy, which cannot reliably distinguish between pathogenic and non-pathogenic species. It is equally important to address structural inequalities, especially by expanding access to safe water and sanitation in underserved regions such as rural areas of Sabah and Sarawak. In addition, integrating protozoa-focused education into maternal and child health programmes, along with opportunistic screening in HIV clinics, would help protect vulnerable groups. Collectively, these targeted interventions can ensure that epidemiological findings are translated into meaningful action for both clinical management and public health.

Furthermore, it is necessary to emphasize the strengths of this study. Our work provides valuable insights into the prevalence and geographical distribution of major intestinal protozoa, including *Entamoeba spp*., *G. lamblia* and *Cryptosporidium spp*., and provides a comprehensive overview of the prevalence of IPI in Malaysia. A large number of studies were included in this review, which increases the robustness of the pooled estimates and allows for a more reliable analysis of trends over time. In addition, by assessing multiple risk factors such as socioeconomic status, hygiene practices and access to clean water, this study provides a more holistic understanding of the factors influencing IPI transmission. Another strength lies in the inclusion of different diagnostic methods, which allows for comparative analysis and highlights the need for improved detection strategies. The observed high heterogeneity indicates considerable differences between the studies, probably due to differences in design, population, setting and diagnostic approaches. To address this, a random effects model was applied, and a subgroup analysis based on diagnostic methods showed consistently higher prevalence in studies that used multiple methods. This suggests that reliance on a single diagnostic method may underestimate the true burden of infection.

Therefore, although the pooled estimates remain informative, they should be interpreted with caution. Standardized diagnostic protocols are recommended for future studies to reduce variability and improve comparability. Despite these methodological differences, this review helps to identify important knowledge gaps and areas for future research to ultimately support more effective surveillance and intervention strategies for intestinal protozoal infections in Malaysia.

## 5. Conclusion

This review provides an overview of the epidemiology of intestinal protozoa in Malaysia. The overall prevalence of IPI was 24%. These studies provide useful information for the development of a successful evidence-based public health policy in the context of the Malaysian population to reduce the impact of intestinal protozoal infections. Specific policy measures could be implemented such as introducing routine stool screening in indigenous settlements and rural schools, strengthening diagnostic capacity in district hospitals with affordable molecular tools, and expanding access to clean water and sanitation in underserved regions. Integrating protozoa-focused health education into maternal and child health programmes, and opportunistic screening of high-risk groups such as HIV patients, would further protect vulnerable populations. Embedding these strategies into Malaysia's national parasitic disease control framework could significantly reduce the long-term public health impact of intestinal protozoal infections.

### 5.1. Limitations of the study

This review has several limitations with significant differences in study design, diagnostic techniques and population characteristics are reflected in the high heterogeneity ($I^2 > 98\%$) of the included studies. The widespread use of microscopy,

which was used in more than half of the studies, may have led to an overestimation of prevalence. These factors highlight the need for more representative sampling and the use of standardized, sensitive diagnostic tools in future studies. Possible publication bias was analysed using the funnel plot and the Egger test. Although some asymmetry was observed, the overall impact on the pooled estimates is probably minimal due to the relatively large number of studies and the consistency of the results in the subgroup analyses. Nevertheless, the possibility of publication bias cannot be completely ruled out and should be considered when interpreting the results. Although none of the included studies were categorised as high risk of bias using the Joanna Briggs Institute tool, some of them were of moderate quality. This may have influenced the accuracy of the prevalence estimates and contributed to the observed heterogeneity. In addition, most of the data come from West Malaysia and only a small proportion from other regions, particularly East Malaysia. The results should be interpreted with caution as these factors may affect the generalisability of the pooled prevalence estimates to the entire Malaysian population.

## Supporting information

**S1 Appendix.  Search strategy and sample search terms.**
(DOCX)

**S2 Appendix.  Included studies.**
(DOCX)

**S3 Appendix.  Excluded studies and reasons for exclusion.**
(DOCX)

**S4 Appendix.  Sub-group forest plot.**
(DOCX)

**S5 Appendix.  Risk of bias assessment.**
(DOCX)

**S6 Appendix.  Risk bias assessment results.**
(DOCX)

**S7 Appendix.  Sensitivity analysis forest plot after removing outlier.**
(DOCX).

**S8 Appendix.  Multivariable meta-regression model based on IPI estimates from pooled prevalence rate cases in Malaysia.**
(DOCX)

**S1 File.  PRISMA 2020 checklist.**
(DOCX)

## Acknowledgments

The author wishes to express gratitude to the Institute of Medical Molecular Biotechnology (IMMB), Faculty of Medicine and Universiti Teknologi MARA (UiTM), for their tremendous assistance.

## Author contributions

**Conceptualization:** Hassanain Al-Talib, Nor Shazlina Mizan.

**Data curation:** Nor Shazlina Mizan, Hasnah Ma'amor.

**Formal analysis:** Hassanain Al-Talib, Nor Shazlina Mizan, Hasnah Ma'amor, Seok Mui Wang.

**Methodology:** Hassanain Al-Talib, Nor Shazlina Mizan.

**Supervision:** Hassanain Al-Talib, Seok Mui Wang.

**Validation:** Seok Mui Wang.

**Visualization:** Nor Shazlina Mizan, Hasnah Ma'amor.

**Writing – original draft:** Nor Shazlina Mizan.

**Writing – review & editing:** Hassanain Al-Talib, Nor Shazlina Mizan, Hasnah Ma'amor, Seok Mui Wang.

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
