## [Decision Letter · Decision Letter 0]

15 May 2025

PONE-D-25-15079Prevalence and Risk Factors of Intestinal Protozoal Infections Among Patients in Malaysia: A Systematic Review and Meta-AnalysisPLOS ONE

Dear Dr. Al-Talib,

Thank you for submitting your manuscript to PLOS ONE. After careful consideration, we feel that it has merit but does not fully meet PLOS ONE’s publication criteria as it currently stands. Therefore, we invite you to submit a revised version of the manuscript that addresses the points raised during the review process.

We look forward to receiving your revised manuscript.

Kind regards,

Masoud Foroutan, Ph.D; Assistant Professor

Academic Editor

PLOS ONE

 [This research received fund from Universiti Teknologi MARA (UiTM), 600-RMC/PRGS 5/3 (004/2024)]. 

3. As required by our policy on Data Availability, please ensure your manuscript or supplementary information includes the following:

Additional Editor Comments (if provided):

Reviewers' comments:

Reviewer's Responses to Questions

**Comments to the Author**

1. Is the manuscript technically sound, and do the data support the conclusions?

Reviewer #1: Yes

Reviewer #2: Partly

Reviewer #3: Yes

2. Has the statistical analysis been performed appropriately and rigorously? 

Reviewer #1: Yes

Reviewer #2: No

Reviewer #3: No

3. Have the authors made all data underlying the findings in their manuscript fully available?

Reviewer #1: Yes

Reviewer #2: No

Reviewer #3: Yes

4. Is the manuscript presented in an intelligible fashion and written in standard English?

Reviewer #1: Yes

Reviewer #2: No

Reviewer #3: No

5. Review Comments to the Author

Reviewer #1: I would like to express my sincere appreciation for the opportunity to review this valuable and comprehensive work. The manuscript presents an insightful and systematic meta-analysis of intestinal protozoal infections (IPIs) in Malaysia, with a strong methodological framework and a high quality of included studies.

I have some minor questions:

1. What methodological approach was used to assess the overall quality of the studies, and how did it categorize the studies based on their scores?

2. How did the different diagnostic methods (microscopy, molecular, immunoassay, and combinations thereof) impact the reported prevalence rates of intestinal protozoal infections?

3. What does the high heterogeneity index (I²) reported in the meta-analysis suggest about the included studies, and how might this affect the interpretation of the pooled prevalence rates?

4. According to the article, how did the prevalence of intestinal protozoal infections differ across geographical regions such as Perak, Selangor, and Pahang, and what might explain these variations?

5. Discuss the significance of the Egger’s test result in evaluating the presence of publication bias. How does this result align with the visual interpretation of the funnel plot?

6. What socioeconomic and hygiene-related risk factors were most strongly associated with increased intestinal protozoal infections, and how did the authors support these conclusions?

7. In the article, infectious agents such as Entamoeba spp., Giardia lamblia, and Cryptosporidium spp. are mentioned. How does the prevalence of these species compare, and how should they be formatted when writing about them? (Note: In your manuscript, make sure to italicize the names of organisms.)

8. How does the sensitivity analysis strengthen the reliability of the meta-analysis findings, particularly in light of the identified publication bias?

Reviewer #2: General comments

The manuscript presents a systematic review and meta-analysis on the prevalence and risk factors of intestinal protozoal infections (IPI) in Malaysia. While the study addresses an important public health issue and follows PRISMA guidelines, several critical issues must be addressed to strengthen its validity and impact.

1. Critical concerns: High Heterogeneity (I²=98.22%): The extreme heterogeneity undermines the reliability of pooled estimates. The authors should perform meta-regression to identify sources of variability (e.g., regional differences, diagnostic methods, population characteristics).

2. Publication bias: Egger’s test confirms significant bias. A sensitivity analysis (e.g., the trim-and-fill method) should be conducted to assess the impact of missing studies.

3. Missing supplementary materials: Figures (e.g., PRISMA flowchart, forest plots) and supplementary tables are referenced but not provided, hindering transparency and reproducibility. These must be included.

4. Diagnostic limitations: Overreliance on microscopy (28/51 studies) likely underestimates species-specific prevalence (e.g., inability to distinguish Entamoeba histolytica from E. dispar). This limitation should be explicitly discussed, and its impact on results quantified.

5. Regional disparities: Peninsular Malaysiaa is over-represented (18 studies) vs. East Malaysia (1 study). This imbalance limits generalisability. The authors should acknowledge this and discuss its implications.

6. Language bias: Excluding non-English studies may omit locally relevant data. Future reviews should incorporate multilingual databases to mitigate this.

7. Discussion enhancements: Please expand on how diagnostic method variability affects prevalence estimates. Again, discuss strategies to improve IPI surveillance (e.g., standardised molecular diagnostics).

8. Limitations section: Explicitly state how high heterogeneity, regional disparities, and diagnostic limitations impact conclusions.

9. In general, it is an extension of your previous published work. Mizan, N. S., Al-Talib, H., & Wang, S. M. (2025). Epidemiological Characteristics of Intestinal Protozoal Infections and Their Risk Factors in Malaysia: Systematic Review and Meta-Analysis Protocol. JMIR Research Protocols, 14(1), e66350.

https://doi.org/10.2196/66350

PMid:40184188 PMCid:PMC12008696

Reviewer #3: Dear Authors,

Thank you for your submission. The topic of intestinal protozoal infections is relevant and timely. Your adherence to PRISMA guidelines and comprehensive data extraction are commendable. However, several major revisions are necessary before publication:

The title suggests a focus on both prevalence and risk factors, but the latter is addressed only descriptively. Please consider a formal meta-analysis of risk estimates (e.g., OR, RR) or revise the title accordingly.

Include the full search strategy used for each database in the supplementary materials.

Present a summary of JBI scores and classify the quality of studies (high/moderate/low).

Explore heterogeneity more robustly with expanded meta-regression and sensitivity analysis.

Correct for publication bias using appropriate methods (e.g., Trim and Fill).

Address minor language and structural inconsistencies.

With these improvements, the manuscript has strong potential for publication.

6. PLOS authors have the option to publish the peer review history of their article (what does this mean? ). If published, this will include your full peer review and any attached files.

**Do you want your identity to be public for this peer review?** For information about this choice, including consent withdrawal, please see our Privacy Policy .

Reviewer #1: **Yes: ** Milad Badri

Reviewer #2: No

Reviewer #3: No

---

## [Author Response · Author response to Decision Letter 1]

19 Jul 2025

Summary of Editor Comments and Responses

The editor first requested clarification on the role of funders. In response, we have included the statement, “The funders had no role in study design, data collection and analysis, decision to publish, or preparation of the manuscript,” in the cover letter, as instructed.

Regarding the journal’s Data Availability policy, we have provided a numbered table of all studies identified during the literature search, including those excluded from the analysis along with reasons for their exclusion. This table is included in the supplementary files (S2 and S3). We have also uploaded a full data extraction table that contains the names of data extractors, dates of extraction, confirmation of study eligibility, and all extracted data required to replicate our systematic review and/or meta-analysis. This file is included as “meta_RAWDATA.” Additionally, we used the Joanna Briggs Institute (JBI) critical appraisal tool to assess the risk of bias across all included studies. The completed risk of bias table is included in the supplementary files (S6). Details regarding the handling of missing data are explained in the Data Extraction section of the manuscript (Page 6, highlighted in yellow), and all relevant risk of bias and data handling information are provided both in the main text and supplementary materials.

Reviewer 1 highlighted several areas for improvement. In response to their first comment, we clarified that the JBI critical appraisal tools were used to assess study quality, as described in the Quality Assessment section on Pages 6–7. We also addressed how different diagnostic methods impacted the reported prevalence in the Discussion section (Page 21). The significance of the high heterogeneity index (I²) is explained in the Data Synthesis section (Pages 7–8), and regional variations in prevalence (e.g., Perak, Selangor, Pahang) are discussed on Page 22. We addressed publication bias through Egger’s test, with results discussed on Page 20. Socioeconomic and hygiene-related risk factors are discussed on Page 13. The comparison of species-specific prevalence is detailed on Page 23, and all species names (e.g., Entamoeba spp., Giardia lamblia, Cryptosporidium spp.) are formatted according to scientific conventions (italicized with appropriate capitalization). Sensitivity analysis findings are presented in the Discussion section on Page 26.

Reviewer 2 raised concerns about heterogeneity, publication bias, and missing supplementary materials. In response, we conducted a meta-regression analysis (results on Page 20; Supplementary S8) and a trim-and-fill sensitivity analysis (Page 19). All missing supplementary materials, including the PRISMA flowchart, forest plots, and tables, have now been included in the revised submission. We also discussed the limitations of microscopy-based diagnostics on Page 21 and acknowledged the regional disparity in study representation (Page 22). We clarified that language bias was mitigated by including both English and Bahasa Malaysia articles during screening. Strategies to improve surveillance (such as standardizing molecular diagnostics) are discussed in the Discussion section (Pages 22–23), and limitations including high heterogeneity and diagnostic issues are stated on Page 27. Additionally, we clarified how this manuscript builds upon our previously published protocol in the Introduction (Page 5).

Reviewer 3 suggested improving the depth of risk factor analysis and methodological transparency. In response, we included a formal meta-analysis of risk factor estimates rather than presenting them descriptively. The complete search strategy for each database is now provided in Supplementary File S1. A summary of the JBI scores and classification of study quality (high/moderate/low) is included in Table 3, with full details in Supplementary File S6. We expanded our heterogeneity assessment and performed additional meta-regression and sensitivity analyses (Pages 19–20). The trim-and-fill method was applied as part of the publication bias assessment (Page 19). Finally, we revised the manuscript for minor language and structural issues to improve clarity and consistency.

---

## [Decision Letter · Decision Letter 1]

6 Aug 2025

PONE-D-25-15079R1Prevalence and Risk Factors of Intestinal Protozoal Infections Among Patients in Malaysia: A Systematic Review and Meta-AnalysisPLOS ONE

Dear Dr. Al-Talib,

Thank you for submitting your manuscript to PLOS ONE. After careful consideration, we feel that it has merit but does not fully meet PLOS ONE’s publication criteria as it currently stands. Therefore, we invite you to submit a revised version of the manuscript that addresses the points raised during the review process.

We look forward to receiving your revised manuscript.

Kind regards,

Masoud Foroutan, Ph.D; Assistant Professor

Academic Editor

PLOS ONE

Journal Requirements:

Reviewers' comments:

Reviewer's Responses to Questions

**Comments to the Author**

1. If the authors have adequately addressed your comments raised in a previous round of review and you feel that this manuscript is now acceptable for publication, you may indicate that here to bypass the “Comments to the Author” section, enter your conflict of interest statement in the “Confidential to Editor” section, and submit your "Accept" recommendation.

Reviewer #1: All comments have been addressed

Reviewer #2: All comments have been addressed

2. Is the manuscript technically sound, and do the data support the conclusions?

Reviewer #1: Yes

Reviewer #2: Yes

3. Has the statistical analysis been performed appropriately and rigorously? 

Reviewer #1: Yes

Reviewer #2: Yes

4. Have the authors made all data underlying the findings in their manuscript fully available?

Reviewer #1: Yes

Reviewer #2: Yes

5. Is the manuscript presented in an intelligible fashion and written in standard English?

Reviewer #1: Yes

Reviewer #2: Yes

6. Review Comments to the Author

Reviewer #1: The authors have answered all of my questions and the paper has been greatly improved. Therefore,

it can be accepted for publication..........

Reviewer #2: General Comments:

The manuscript titled "Prevalence and Risk Factors of Intestinal Protozoal Infections Among Patients in Malaysia: A Systematic Review and Meta-Analysis" provides a comprehensive and well-structured analysis of intestinal protozoal infections (IPI) in Malaysia. The study addresses a significant public health issue and fills a critical gap in the literature by synthesizing data from 49 studies. The methodology is robust, adhering to PRISMA guidelines, and the results are presented clearly with appropriate subgroup analyses. However, minor revisions are required to enhance clarity, address methodological limitations, and ensure the manuscript meets the journal's standards.

1. Abstract: The abstract is concise and summarizes the key findings effectively. However, consider adding a sentence about the clinical or public health implications of the 24% pooled prevalence to highlight the study's significance.

2. Introduction: Please consider briefly mentioning the global burden of IPI to contextualize the Malaysian data.

3. Methods: Clarify why studies from East Malaysia were excluded due to a focus on animal/environmental samples. This could introduce bias, and a brief justification would strengthen the manuscript. Again, the statistical methods are appropriate, but the high heterogeneity (I² > 98%) should be discussed earlier in the results section to prepare readers for its implications.

4. Results: Consider adding a brief summary of the risk factors in the main text to complement Table 5.

5. Discussion: Please strengthen the section on limitations by explicitly stating how the high heterogeneity and regional disparities might affect the generalizability of the findings. Again, the public health recommendations (e.g., improved water access, hygiene education) are relevant but could be expanded with specific policy suggestions.

6. Tables and Figures: Ensure that the supplementary files (e.g., PRISMA flowchart, forest plots) are referenced in the main text where appropriate.

7. References: Check for consistency in formatting (e.g., italics for species names, journal abbreviations).

7. PLOS authors have the option to publish the peer review history of their article (what does this mean? ). If published, this will include your full peer review and any attached files.

**Do you want your identity to be public for this peer review?** For information about this choice, including consent withdrawal, please see our Privacy Policy .

Reviewer #1: **Yes: ** ------------

Reviewer #2: No

---

## [Author Response · Author response to Decision Letter 2]

14 Aug 2025

We thank the reviewer for the constructive feedback, which has helped us improve the clarity and quality of the manuscript.

In the Abstract, we have added a sentence highlighting the clinical and public health implications of the 24% pooled prevalence, emphasizing its significance for disease control and prevention strategies (highlighted in yellow on page 3). The added sentence reads: “The high pooled prevalence of 24% underscores a ……. reducing prevalence and transmission.”

In the Introduction, we have included a sentence on the global burden of intestinal protozoal infections (IPI) to contextualize the Malaysian data (highlighted in yellow on page 4): “Given that IPI remain a significant … local share of this worldwide burden. [2].”

In the Methods section, we have added a justification explaining that studies from East Malaysia were excluded because they primarily focused on animal or environmental samples, which fell outside the scope of this review targeting human infection prevalence. We have also acknowledged that this exclusion could introduce bias and noted it as a limitation (highlighted in yellow on page 6): “Most East Malaysian studies …… fell outside our scope.” In response to this comment, we have also added one additional reference and updated the reference list accordingly (highlighted in green on page 37).

In the Results section, we have added a brief summary of the risk factors in the main text to complement Table 5 (highlighted in yellow on page 11): “A subgroup meta-analysis revealed a higher prevalence among …… eating raw vegetables (40%).” Additionally, we have addressed the suggestion regarding heterogeneity by mentioning the high I² value (>98%) earlier in the Results section, immediately after reporting the pooled prevalence, along with a brief explanation of potential sources of heterogeneity (highlighted in yellow on page 9): “A high level of heterogeneity… diagnostic methods and study designs.”

In the Discussion, we have strengthened the Limitations section by explicitly stating how high heterogeneity and regional disparities may affect the generalizability of the pooled prevalence estimates (highlighted in yellow on page 27): “In addition, most of …… to the entire Malaysian population.” We have also expanded the public health recommendations by adding specific policy suggestions (highlighted in yellow on page 27): “Specific policy measures could be implemented such as …… infections on public health could be steadily reduced.”

For the Tables and Figures, we have ensured that all supplementary files (e.g., PRISMA flowchart, forest plots) are appropriately referenced in the main text. These references are highlighted in green in the relevant sections: S1 (page 5), S2 (page 6), S3 (page 6), S4 (page 10), S5 (page 7), S6 (page 8), S7 (page 12), S8 (page 12), and S9 (page 5).

Finally, regarding the References, we have checked for consistency in formatting, ensuring that all species names are italicized and that journal abbreviations follow the journal’s style guidelines.

---

## [Decision Letter · Decision Letter 2]

18 Aug 2025

PONE-D-25-15079R2Prevalence and Risk Factors of Intestinal Protozoal Infections Among Patients in Malaysia: A Systematic Review and Meta-AnalysisPLOS ONE

Dear Dr. Al-Talib,

Thank you for submitting your manuscript to PLOS ONE. After careful consideration, we feel that it has merit but does not fully meet PLOS ONE’s publication criteria as it currently stands. Therefore, we invite you to submit a revised version of the manuscript that addresses the points raised during the review process.

**ACADEMIC EDITOR: **

We look forward to receiving your revised manuscript.

Kind regards,

Masoud Foroutan, Ph.D; Assistant Professor

Academic Editor

PLOS ONE

Journal Requirements:

Reviewers' comments:

Reviewer's Responses to Questions

**Comments to the Author**

1. If the authors have adequately addressed your comments raised in a previous round of review and you feel that this manuscript is now acceptable for publication, you may indicate that here to bypass the “Comments to the Author” section, enter your conflict of interest statement in the “Confidential to Editor” section, and submit your "Accept" recommendation.

Reviewer #2: (No Response)

2. Is the manuscript technically sound, and do the data support the conclusions?

Reviewer #2: Yes

3. Has the statistical analysis been performed appropriately and rigorously? 

Reviewer #2: Yes

4. Have the authors made all data underlying the findings in their manuscript fully available?

Reviewer #2: Yes

5. Is the manuscript presented in an intelligible fashion and written in standard English?

Reviewer #2: Yes

6. Review Comments to the Author

Reviewer #2: General Comments:

The manuscript titled "Prevalence and Risk Factors of Intestinal Protozoal Infections Among Patients in Malaysia: A Systematic Review and Meta-Analysis" provides a comprehensive and well-structured analysis of intestinal protozoal infections (IPI) in Malaysia. The study addresses an important public health issue and offers valuable insights into the epidemiology, risk factors, and regional variations of IPI. The methodology is robust, adhering to PRISMA guidelines, and the results are clearly presented. However, some minor revisions are required to enhance clarity, address methodological limitations, and ensure the manuscript meets the journal's standards.

1. Abstract: The abstract is concise and informative but could briefly mention the high heterogeneity observed (I² > 98%) and its implications for interpreting the pooled prevalence estimates.

2. Introduction: The background effectively contextualizes the study, but the transition to the Malaysian context could be smoother. Consider adding a sentence or two about why Malaysia is a critical region for studying IPI (e.g., diverse populations, urbanization vs. rural disparities, etc.).

3. Methods: Search Strategy: The inclusion of multiple databases is commendable. However, the exclusion of East Malaysian studies due to their focus on animal/environmental samples should be explicitly justified in the main text (not just in the response to reviewers) to avoid potential bias. Again, Quality Assessment: The Joanna Briggs Institute tool is appropriate, but the results of the quality assessment (e.g., high risk of bias in some studies) should be discussed in the limitations section.

4. Results: Subgroup Analyses: The regional and population-based subgroup analyses are well-presented. However, the high heterogeneity (I² > 98%) suggests significant variability across studies. The authors should discuss potential reasons (e.g., differences in diagnostic methods, study populations) and how this affects the interpretation of pooled estimates. Again, Risk Factors: The meta-analysis of risk factors is a strength, but the results should be interpreted cautiously due to residual confounding. Consider adding a sentence about this limitation.

5. Discussion: Clinical and Public Health Implications: The discussion effectively highlights the need for interventions (e.g., improved sanitation, hygiene education). However, it could be strengthened by proposing specific, actionable recommendations for policymakers (e.g., targeted screening in high-risk groups like indigenous communities).

6. Limitations: The limitations section is thorough but should explicitly mention the potential for publication bias (as indicated by the funnel plot and Egger test) and its impact on the results.

7. Tables and Figures: Table 3: Ensure all abbreviations (e.g., "Ref") are defined in the footnote. Again, Figure 6 (Funnel Plot): The asymmetry suggests possible publication bias. This should be discussed in the limitations.

7. References: The references are comprehensive and up-to-date. However, check for consistency in formatting (e.g., italicization of species names, journal abbreviations).

8. Supplementary materials: Ensure all supplementary files (e.g., PRISMA flowchart, forest plots) are referenced in the main text and uploaded correctly.

7. PLOS authors have the option to publish the peer review history of their article (what does this mean? ). If published, this will include your full peer review and any attached files.

**Do you want your identity to be public for this peer review?** For information about this choice, including consent withdrawal, please see our Privacy Policy .

Reviewer #2: No

---

## [Author Response · Author response to Decision Letter 3]

27 Aug 2025

We thank the reviewers for their constructive comments and suggestions, which have greatly helped to improve the manuscript. In the abstract, we have revised the text to briefly mention the high heterogeneity (I² > 98%) and its implications for interpreting the pooled prevalence estimates (page 3, highlighted in green). In the introduction, we strengthened the transition to the Malaysian context by highlighting why Malaysia is an important setting for studying intestinal protozoal infections, considering its diverse populations and rural–urban disparities (page 4, highlighted in green). In the methods section, we have explicitly stated the rationale for excluding East Malaysian studies that focused on animal and environmental samples (page 6, highlighted in green). Regarding quality assessment, while none of the included studies were rated as high risk, several were of moderate quality; this has been acknowledged in the limitations section as a factor that may have influenced the pooled prevalence estimates and contributed to heterogeneity (page 28, highlighted in yellow).

In the results, we have expanded the discussion of subgroup analyses to address the high heterogeneity (I² > 98%) and possible sources such as differences in diagnostic methods and study populations, noting that pooled estimates should be interpreted with caution (page 24, highlighted in green). We also revised the section on risk factors to emphasize that the findings should be interpreted cautiously due to potential residual confounding (page 12, highlighted in green). In the discussion, we have strengthened the clinical and public health implications by adding specific, actionable recommendations for policymakers, such as targeted screening in high-risk groups including indigenous communities (page 26, highlighted in green).

The limitations section has been expanded to acknowledge the possibility of publication bias, as indicated by the funnel plot and Egger test, in addition to the inclusion of studies of moderate methodological quality (page 28, highlighted in green). For tables and figures, we clarified that the abbreviation “Ref” appears in Table 7 (not Table 3) and revised the table footnote to define all abbreviations. We also acknowledged the asymmetry in the funnel plot (Figure 6) and discussed its implication of potential publication bias in the limitations section (page 28, highlighted in green). References have been carefully reviewed for consistency, including italicization of species names and standardization of journal abbreviations. Finally, all supplementary files, including the PRISMA flowchart and forest plots, have been properly referenced in the main text and uploaded as required.

---

## [Editor Report · Decision Letter 3]

28 Aug 2025

Prevalence and Risk Factors of Intestinal Protozoal Infections Among Patients in Malaysia: A Systematic Review and Meta-Analysis

PONE-D-25-15079R3

Dear Dr. Al-Talib,

We’re pleased to inform you that your manuscript has been judged scientifically suitable for publication and will be formally accepted for publication once it meets all outstanding technical requirements.

Kind regards,

Masoud Foroutan, Ph.D; Assistant Professor

Academic Editor

PLOS ONE
---

## [Editor Report · Acceptance letter]

PONE-D-25-15079R3

PLOS ONE

Dear Dr. Al-Talib,

I'm pleased to inform you that your manuscript has been deemed suitable for publication in PLOS ONE. Congratulations! Your manuscript is now being handed over to our production team.

Kind regards,

on behalf of

Dr. Masoud Foroutan

Academic Editor

PLOS ONE